# Uniform Approximation of Equivariant/Invariant Neural Networks

## Abstract

Equivariant structures have been widely adopted in graph neural networks due to their demonstrated effectiveness in various machine learning tasks, such as point clouds, biology, and chemistry. We focuses on investigating the approximation power of equivariant neural networks. Specifically, we prove that equivariant neural networks with any continuous activation function can approximate any continuous equivariant function. Our theorem is established based on a novel composition of any subgroup $G$ of a symmetric group $S_M$. Additionally, we note that this representation may not work for certain invariant continuous functions when the dimension of the latent space is smaller than the dimension of the input space.

## 1 Introduction

Deep learning has achieved wide application in many fields such as image recognition Gao et al. (2019) and extreme weather event prediction Chattopadhyay et al. (2020), but it usually requires a lot of data. However, some data is inherently very expensive and difficult to acquire, such as medical images in Ma et al. (2020). To utilize data more efficiently, Vinyals et al. (2015) proposed a basic setup of learning from unordered sets, and research in this area has received increasing attention in recent years. This setup mainly considers the case where set elements are represented as feature vectors, with little emphasis on set elements themselves obeying their own symmetries. Building on this, Cohen & Welling (2016) proposed a framework of permutations equivariant and permutation invariant structures, which generalizes convolutional neural networks with invariance to permutations, greatly enhancing the expressive power without increasing parameter count. This indicates equivariant neural networks have many excellent properties, but their approximation abilities need further investigation. Therefore, we needs to explore the approximation abilities of equivariant neural networks.

Equivariant neural networks have many different architectures and applications, such as pretraining on chemical data with faster convergence and lower loss values based on tensor field networks for end-to-end message passing neural network TSNNetJackson et al. (2021), equivariant neural networks built from the Lorentz group based on finite-dimensional representation theory that perform remarkably on classification tasks in particle physicsBogatskiy et al. (2020), neural network architectures for rotation and permutation equivariance on 2D point cloud dataBökman et al. (2022), and SE(3)-Transformer variants with self-attention modules for 3D point clouds and graphsFuchs et al. (2020), see related work for details. These articles on applications of equivariant neural networks demonstrate the powerful working ability and generalization ability of equivariant neural networks. From a theoretical perspective, it can be seen that TSNNet is a $SO(3) \times S_N$-equivariant point cloud neural network; Lorentz group equivariant neural network is a Lorentz group-equivariant neural network, ZZNet is an equivariant neural network based on general group/semigroup, and $SE(3)$-Transformer is an equivariant neural network that is equivariant under 3D rotations and translations. It can be seen that **general groups** are important for equivariant neural networks. Therefore, we mainly focuses on studying the approximation theory of equivariant neural networks on general groups.

First, we discusses a special case in equivariant neural networks, invariant neural networks. According to the discussion in Zaheer et al. (2017) (which established the DeepSets network architecture), any $S_M$-invariant neural network can approximate $S_M$-invariant continuous functions. In addi-

tion, Zaheer et al. (2017) also proposed a permutation invariant continuous function representation, namely,

$$f(\boldsymbol{x}_1, \boldsymbol{x}_2, \cdots, \boldsymbol{x}_M) = \rho\left(\sum_{i=1}^{M} \phi(\boldsymbol{x}_i)\right),$$

Figure 1 is a powerful tool for understanding this conclusion. After Zaheer et al. (2017), Maron et al. (2019) provided theoretical guarantees for uniform approximation of $G$-invariant functions using the Stone-Weierstrass theorem, where $G$ is any subgroup of the permutation group $S_M$. However, Maron et al. (2019) only proved the necessary conditions for uniform approximation by $G$-invariant networks containing first-order tensors.

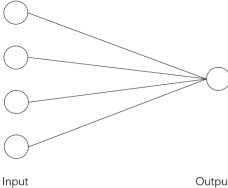

Figure 1: Illustration of a permutation invariant neural network layer, same color indicates identical weights.

Secondly, in terms of the development of equivariant neural networks, according to the discussion in Zaheer et al. (2017), the function $\sigma(\theta\boldsymbol{x})$ is permutation equivariant if and only if the off-diagonal elements of the matrix $\theta \in \mathbb{R}^{M \times M}$ are tied together and the diagonal elements are equal, that is,

$$\theta = \lambda\mathbf{I} + \gamma\left(11^{\top}\right) \quad \lambda, \gamma \in \mathbb{R}, 1 = [1, \ldots, 1]^{\top} \in \mathbb{R}^{M},$$

Where $\mathbf{I} \in \mathbb{R}^{M \times M}$ represents the identity matrix. Figure 2 can help understand this concept. Nevertheless, Zaheer et al. (2017) was unable to provide a corresponding representation form for any permutation equivariant function $F : R^M \to R^M$. Fortunately, Zaheer et al. (2017) provided a general method to use $S_M$-equivariant neural networks to approximate any $S_M$-equivariant continuous function $F : \mathbb{R}^M \to \mathbb{R}^M$. However, this method does not provide theoretical guarantees for the approximation and representation of $G$-equivariant continuous functions.

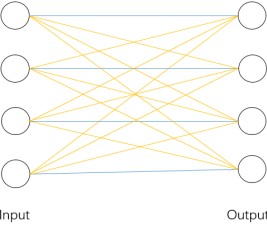

Figure 2: Illustration of a permutation equivariant neural network layer, same color indicates identical weights.

After that, Segol & Lipman (2019) extended the basic model DeepSets in Zaheer et al. (2017) to general groups, and using deep neural networks that are permutation invariant or equivariant to learn functions on unordered sets has become commonplace. Yarotsky (2022) proved the universal approximator for continuous $SE(2)$-equivariant signal transformations by constructing a complete invariant/equivariant network with intermediate polynomial layers. Lawrence (2022) provided not only invariant but also effective approximation results by defining an invariant approximation architecture with novel invariant nonlinearities that capture invariant function smoothness.

In this paper, we aim to address the aforementioned issues and provide concrete solutions. Specifically, given any natural numbers $M$ and $N$ and $\mathcal{X} = \mathbb{R}^3$, consider a $SO(3) \times G$-invariant continuous function $f$ defined on $\mathcal{X}^M$ to $\mathbb{R}^N$. Our goal is to provide a concrete representation form so that the function can be accurately characterized. Inspired by previous work Zaheer et al. (2017), especially for the case when $G = S_M$, we will adopt the approach therein to represent $f$,

$$\phi(\boldsymbol{x}) = \left( \|\boldsymbol{x}\|^l Y_{\ell m} \left( \frac{\boldsymbol{x}}{\|\boldsymbol{x}\|} \right) \right)_{m \in \{-l, \cdots, l\}}.$$

Through the above discussion, we reach our first conclusion: $SO(3) \times G$-invariant neural networks can approximate any $SO(3) \times G$-invariant continuous function. And we provide a compositional representation of $SO(3) \times G$-invariant continuous functions in terms of component continuous functions and spherical harmonics.

Inspired by Sannai et al. (2019), we connect $S_M$-equivariant continuous functions with $\text{Stab}_{S_M}(k)$-invariant continuous functions. We upgrade $S_M$ to a more general group $G$. Therefore, we can obtain concrete continuous representations of $G$-equivariant and $SO(3)$-invariant continuous functions by composing continuous functions with spherical harmonics; and we generalize the concept of the neural network form $\sigma(\theta \boldsymbol{x})$ from Zaheer et al. (2017) to more general equivariant functions $F(\boldsymbol{x})$. In summary, we can draw the second conclusion: $G$-equivariant and $SO(3)$-invariant neural networks can approximate any $G$-equivariant and $SO(3)$-invariant continuous function. And the above conclusions do not require the invertibility of the activation function $\sigma(\boldsymbol{x})$.

## 1.1 RELATED WORK

**Theoretical Guarantees for Neural Networks**: A three-layer neural network can be represented using bounded, continuous, monotone, and differentiable sigmoidal activation functions Funahashi & Nakamura (1993); Hornik et al. (1989); Leshno et al. (1993); Cybenko (1989). This conclusion extends to ReLU, Tanh, and other nonlinear activations Funahashi (1989); Hornik (1993); Sonoda & Murata (2017); DeVore et al. (2021). Barron (1994) proposed an upper bound on the integrated square error Barron (1994).

**Equivariant Neural Networks**: The symmetric group $S_M$ is explained in detail Dummit & Foote (2004). Permutation invariance and equivariance are described as actions of $S_M$ Zaheer et al. (2017). Fundamentals are provided in Esteves (2020). Convolutional networks exploiting larger equivalence groups are discussed in Cohen & Welling (2016). Fourier analysis is covered in Chirikjian (2000). Advances in deep learning for point clouds are reviewed in Guo et al. (2020). Canonical group representations are explained in Chirikjian (2000). HamNet with translation and rotation invariant losses is introduced in Li et al. (2021). PointNet, a permutation-invariant network for point clouds, is presented in Qi et al. (2017). Optimization for local structure capture is discussed in Qi et al.. Rotationally invariant geometric features are used in Zhang et al. (2019). Equivariant neural networks for shape analysis are described in Poulenard & Guibas (2021).

**Universal Approximation of Invariant and Equivariant Neural Networks**: Universal approximation properties are studied Zaheer et al. (2017); Wagstaff et al. (2019; 2021); Maron et al. (2019); Yarotsky (2021). Group representations are introduced in Knapp (2001). Dynamical system flow graphs are studied in Li et al. (2022). A more general point cloud architecture is proposed in Finkelshtein et al. (2022).

**Universal Approximation of Equivariant Neural Networks**: Equivariant neural networks are analyzed Keriven & Peyré (2019); Dym & Maron (2020); Sannai et al. (2019); Kumagai & Sannai (2020).

**Spherical Harmonics**: Spherical harmonics are introduced in Blanco et al. (1997). They are combined with $S_M$-equivariant neural networks Thomas et al. (2018). $SO(3)$-invariant functions are based on spherical harmonics Poulenard et al. (2019). Harmonic networks (H-Nets) are discussed in Worrall et al. (2017). Spherical convolutional networks are presented in Esteves et al. (2018).

## 2 PRELIMINARY

### 2.1 EQUIVARIANT NEURAL NETWORKS

In this section, we introduce equivariance and the architecture of equivariant neural networks. A typical deep neural network comprises $n$ layers with neurons alternating between weight operations $\mathcal{W}_i$ and activation functions $\sigma_i$. Formally, each layer $Z_i$ transforms input $\boldsymbol{x}$ from $\mathbb{R}^{d_i}$ to $\mathbb{R}^{d_{i+1}}$:

$$|Z_i(\boldsymbol{x}) = \sigma_i(\mathcal{W}_i(\boldsymbol{x}) + \boldsymbol{b}_i), \tag{1}$$

where $\boldsymbol{b}_i \in \mathbb{R}^{d_{i+1}}$. The entire network $Y(\boldsymbol{x})$ is the composition of these layers:

$$Y(\boldsymbol{x}) = Z_n \circ Z_{n-1} \circ \ldots \circ Z_1(\boldsymbol{x}). \tag{2}$$

The activation functions must satisfy nonlinearity and continuity properties, as established in Funahashi (1989), Cybenko (1989), and Daubechies et al. (2021) for Sigmoid and ReLU activation functions. Boundedness and continuous differentiability, while common, are not mandatory.

For subgroup $G \leq S_M$ within the symmetric group, we define the action of $SO(3) \times G$ on $\boldsymbol{x} = (\boldsymbol{x}_1, \boldsymbol{x}_2, \ldots, \boldsymbol{x}_M) \in \mathcal{X}^M$ as follows: for any $\alpha \in SO(3)$ and $\sigma \in G$,

$$\alpha \circ \sigma(\alpha \boldsymbol{x}1, \alpha \boldsymbol{x}2, \cdots, \alpha \boldsymbol{x}M) = (\alpha \boldsymbol{x}\sigma^{-1}(1), \alpha \boldsymbol{x}\sigma^{-1}(2), \cdots, \alpha \boldsymbol{x}\sigma^{-1}(M)),$$

where $\boldsymbol{x}_i \in \mathcal{X} \subset \mathbb{R}^3$.

**Definition 1** (*G-invariant function*). *For any $M, N \in \mathbb{N}$, any $\sigma \in G$, and any element $\boldsymbol{x} = (\boldsymbol{x}_1, \boldsymbol{x}_2, \cdots, \boldsymbol{x}_M) \in \mathcal{X}^M$, $\mathcal{X} \subset \mathbb{R}^3$, if the function $f : \mathcal{X}^M \to \mathbb{R}^N$ satisfies the following equation:*

$$f(\sigma \boldsymbol{x}) = f(\boldsymbol{x}),$$

*then the function $f$ is called $G$-invariant.*

The invariant functions $f : \mathcal{X}^M \to \mathbb{R}^N$ examined herein exhibit invariance to the combined action of subgroup $G \leq S_M$ permutations and $SO(3)$ rotations. Formally, for arbitrary $\sigma \in G$ and $\alpha \in SO(3)$, and any input $\boldsymbol{x} \in \mathcal{X}^M$ with $\mathcal{X} \subset \mathbb{R}^3$, the equivalence $F(\alpha \circ \sigma \boldsymbol{x}) \equiv F(\boldsymbol{x})$ holds, encoding joint invariance of $f$ to the $G$ and $SO(3)$ groups.

**Definition 2** (*G-equivariant function*). *For any $M \in \mathbb{N}$, any $\sigma \in G$, and any element $\boldsymbol{x} = (\boldsymbol{x}_1, \boldsymbol{x}_2, \cdots, \boldsymbol{x}_M) \in \mathcal{X}^M$ with $\mathcal{X} \subset \mathbb{R}^3$, if the function $f : \mathcal{X}^M \to \mathbb{R}^M$ satisfies the following equation:*

$$f(\sigma \boldsymbol{x}) = \sigma f(\boldsymbol{x}),$$

*then the function $f$ is called $G$-equivariant.*

The equivariant mappings $F : \mathcal{X}^M \to \mathbb{R}^M$ examined herein exhibit $G$-equivariance and joint $SO(3)$-invariance, satisfying $F(\alpha \circ \sigma \boldsymbol{x}) \equiv \sigma F(\boldsymbol{x}), \forall \sigma \in G, \alpha \in SO(3)$, with $\boldsymbol{x} \in \mathcal{X}^M, \mathcal{X} \subset \mathbb{R}^3$. A justification for imposing $G$-equivariance and $SO(3)$-invariance on the mapping $f$ is provided in Appendix A.2.1. An analysis of full $SO(3)$-equivariance remains an open question for future work.

**Definition 3** (*G-invariant neural network*). *Given a deep neural network as defined in equation 1 with input $\boldsymbol{x} = (\boldsymbol{x}_1, \boldsymbol{x}_2, \cdots, \boldsymbol{x}_M) \in \mathcal{X}^M$, the network is said to be $G$-invariant if and only if its output satisfies*

$$Y(\sigma \boldsymbol{x}) \equiv Y(\boldsymbol{x}), \quad \forall \sigma \in G$$

*where $G$ is a subgroup of the symmetric group $S_M$. That is, the output is unchanged under permutations of the input elements by any $\sigma \in G$.*

Given the existence of positive integer hyperparameters $k, m$ such that $k + 1 \leq m \leq n$, where $n$ is the number of layers, if $\forall \sigma \in G$ each layer mapping $Z_i, 1 \leq i \leq n$ satisfies:

- $Z_i$ is $G$-equivariant, $1 \leq i \leq k$.

- $Z_i$ is $G$-invariant, $k + 1 \leq i \leq m$ then the overall mapping $Y : \mathcal{X}^M \to \mathbb{R}^{d_n+1}$ defined in equation 1 constitutes a $G$-invariant neural network.

**Definition 4** ($G$-equivariant neural network). *Given a deep neural network as defined in equation 1 with input $\boldsymbol{x} = (\boldsymbol{x}_1, \boldsymbol{x}_2, \cdots, \boldsymbol{x}_M) \in \mathcal{X}^M$, the network is said to be $G$-equivariant if and only if its output satisfies:*

$$Y(\sigma \boldsymbol{x}) \equiv \sigma Y(\boldsymbol{x}), \quad \forall \sigma \in G$$

*where $G$ is a subgroup of the symmetric group $S_M$.*

Furthermore, if every layer mapping $Z_i, 1 \leq i \leq n$ is $G$-equivariant, then the overall mapping $Y : \mathcal{X}^M \to \mathbb{R}^{d_n+1}$ in equation 1 constitutes a $G$-equivariant neural network.

### 2.2 APPROXIMATION

The goal of this section is to introduce approximation representations for $G$-invariant neural networks and $G$-equivariant neural networks. To this end, we provide detailed definitions and explanations of uniform approximation below.

**Definition 5** (Uniform approximation). *A sequence of mappings $\{f_n\} : \mathcal{X} \to \mathcal{Y}$ is said to uniformly approximate $f : \mathcal{X} \to \mathcal{Y}$ if and only if for any $\epsilon > 0$ and any $\boldsymbol{x} \in \mathcal{X}$, there exists an integer $N$ such that for all $n > N$,*

$$\max_{\boldsymbol{x} \in \mathcal{X}} |f_n(\boldsymbol{x}) - f(\boldsymbol{x})| < \epsilon.$$

Definition 5 formalizes the notion of a sequence of mappings $f_n : \mathcal{X} \to \mathcal{Y}$ uniformly approximating a target mapping $f : \mathcal{X} \to \mathcal{Y}$. Critically, the choice of $N \in \mathbb{N}$ satisfying the approximation bound is dependent only on the sequence $f_n$ and the approximation precision $\epsilon$, and is invariant to the particular input $\boldsymbol{x} \in \mathcal{X}$. This uniformity over all inputs $\boldsymbol{x} \in \mathcal{X}$ motivates the notion of $f_n$ providing a uniform approximation to $f$ over the domain $\mathcal{X}$.

### 2.3 SPHERICAL HARMONIC FUNCTIONS

Spherical harmonics are special functions defined on the unit ball $\mathbb{S}^2$ in the $\mathbb{R}^3$ space, and can be extended to whole $\mathbb{R}^3$ by homogeneity. Specifically, spherical harmonics originates from solving Laplace's equation in the spherical domains. That is, the general solution $f : \mathbb{R}^3 \to \mathbb{C}$ to Laplace's equation $\Delta f = 0$ in a ball centered at the origin is a linear combination of complex spherical harmonics $Y_l^m$, i.e.,

$$f(r, \theta, \varphi) = \sum_{\ell=0}^{\infty} \sum_{m=-\ell}^{\ell} f_\ell^m r^\ell Y_\ell^m(\theta, \varphi),$$

where $Y_\ell^m(\theta, \varphi)$ is defined as

$$Y_l^m(\theta, \varphi) = N e^{im\varphi} P_l^m(\cos \theta),$$

and $P_l^m$ is an associated Legendre polynomial.

From complex spherical harmonics we can define real spherical harmonics $Y_{\ell m}$ ($l = 0, 1, 2, 3, \cdots$, and $m = -l, -l+1, \cdots, l$) as

$$Y_{\ell m} = \begin{cases} \frac{i}{\sqrt{2}} \left( Y_\ell^m - (-1)^m Y_\ell^{-m} \right) & \text{if } m < 0 \\ Y_\ell^0 & \text{if } m = 0 \\ \frac{1}{\sqrt{2}} \left( Y_\ell^{-m} + (-1)^m Y_\ell^m \right) & \text{if } m > 0 \end{cases}.$$

For a concrete form of spherical harmonics, please refer to Blanco et al. (1997) for details.

We then provide a basic property of real spherical harmonics.

**Property 1** (Orthonormal basis of $\mathcal{L}^2(\mathbb{S}^2)$). *$\{Y_{\ell m}\}_{l \in \mathbb{N}, m \in \{-l, \cdots, l\}}$ is an orthonormal basis of squared-integrable functions on unit sphere $\mathbb{S}^2$, i.e., any $f \in \mathcal{L}^2(\mathbb{S}^2)$, we have*

$$f(\theta, \varphi) = \sum_{l=0}^{\infty} \sum_{m=-l}^{l} f_{\ell m} Y_{\ell m}(\theta, \varphi).$$

**Property 2** (Spherical Harmonics are normalized polynomials). *For any $l \in \mathbb{N}$ and any $m \in \{-l, -l+1, \cdots, l-1, l\}$, $r^l Y_{\ell m}(\theta, \varphi)$ is a $l$-degree polynomial with respect to Cartesian coordinates $(x, y, z)$. Furthermore,*

$$\{r^{l-2k} Y_{(l-2k)m}\}_{k \in \{0, 1, \cdots, \lfloor \frac{l}{2} \rfloor\}, m \in \{2k-l, \cdots, l-2k\}}$$

*forms a basis of $l$-homogeneous polynomials, and*

$$\bigcup_{l=0}^{d} \{r^{l-2k} Y_{(l-2k)m}\}_{k \in \{0, 1, \cdots, \lfloor \frac{l}{2} \rfloor\}, m \in \{2k-l, \cdots, l-2k\}}$$

*forms a basis of polynomials with degree no larger than $d$.*

**Property 3** (Invariant subspace under rotation). *For any fixed $l$, $\mathrm{Span}_{m=-l}^{l}(Y_{\ell m}(x, y, z))$ is an invariant subspace under any rotation. That is, for any rotation transformation $g \in \mathrm{SO}(3)$, $Y_{\ell m}(g(x, y, z)) \in \mathrm{Span}_{m=-l}^{l}(Y_{\ell m}(x, y, z))$.*

The reason is that the spherical harmonic function is invariant under any rotation in Blanco et al. (1997). we can get a $G$-invariant neural network can approximate any $G$-invariant continuous function by the above representation,

$$\phi(\theta \boldsymbol{x}) = D(\theta) \phi(\boldsymbol{x}).$$

where $D(\theta)$ is associated to $\theta$ called the Wigner matrix (of type $l$) and $\boldsymbol{x} \in \mathcal{X} = \mathbb{R}^3$.

# 3 UNIFORM APPROXIMATION OF $G$-INVARIANT NEURAL NETWORKS

This section examines representations for $G$-invariant functions and approximation guarantees for $G$-invariant neural networks. Theorem 1 formally characterizes the $G$-invariant condition and constructs an associated representation in terms of invariant continuous mappings and spherical harmonics. Moreover, the existence of uniform approximation by $G$-invariant networks is established for invariant functions of an arbitrary subgroup $G$. The non-representable scenario is also analyzed, wherein the dimensionality of the latent space precludes representation of functions on the ambient input space, as rigorously codified in Theorem 3. Taken together, these theoretical results provide foundational insights into processing 3D point cloud data with invariance properties.

## 3.1 REPRESENTATION OF $G$-INVARIANT FUNCTIONS

First, we define the **$G$-invariant condition**, which has the following specific representation:

**Definition 6** ($G$-invariant condition). *Consider an arbitrary permutation $\sigma \in G$ where $G$ is a subgroup of the symmetric group $S_M$. A set of scalar coefficients $\{\lambda_i\}_{i=1}^{M}$ is defined to satisfy the $G$-invariant condition if and only if the mapping $\Phi : \mathcal{X}^M \to \mathbb{R}$ given by*

$$\Phi(\boldsymbol{x}) = \sum_{i=1}^{M} \lambda_i \phi(\boldsymbol{x}_i),$$

*satisfies*

$$\Phi(\sigma \boldsymbol{x}) = \Phi(\boldsymbol{x}), \quad \forall \sigma \in G,$$

*where $\phi : \mathcal{X} \to \mathbb{R}^{2l+1}$ denotes the spherical harmonic feature mapping $\phi(\boldsymbol{x}) = \left( |\boldsymbol{x}|^l Y_{lm} \left( \frac{\boldsymbol{x}}{|\boldsymbol{x}|} \right) \right)_{m \in -l, \cdots, l}$.*

This theorem establishes that for arbitrary $G \le S_M$ and continuous $f : \mathcal{X}^M \to \mathbb{R}$ exhibiting joint $SO(3)$ and $G$ invariance on the compact domain $\mathcal{X} \subset \mathbb{R}^3$, there exists a set of coefficients $\{\lambda_i\}_{i=1}^{M}$ satisfying the $G$-invariant condition such that $f$ admits a continuous representation in terms of scalar mapping $\rho : \mathbb{R}^{2l+1} \to \mathbb{R}$ and spherical harmonic basis functions $\phi : \mathcal{X} \to \mathbb{R}^{2l+1}$ with $2l+1 > 3M$.

**Theorem 1.** *Consider $G = \{\sigma_1, \sigma_2, \ldots, \sigma_n\}$ as an arbitrary subgroup of $S_M$, $\mathcal{X}$ is a compact set with $\mathcal{X} \subset \mathbb{R}^3$. If $f : \mathcal{X}^M \to \mathbb{R}$ is a $SO(3) \times G$-invariant continuous function. Then there exists $\{\lambda_i\}_{i=1}^{M} \in \mathbb{R}$ such that $f : \mathcal{X}^M \to \mathbb{R}$ has a continuous representation:*

$$f(\boldsymbol{x}_1, \boldsymbol{x}_2, \cdots, \boldsymbol{x}_M) = \rho\left(\sum_{i=1}^{M} \lambda_i \phi(\boldsymbol{x}_i)\right),$$

where $\rho : \mathbb{R}^{2l+1} \to \mathbb{R}$ is a continuous function, $2l + 1 > 3M$, $\{\lambda_i\}_{i=1}^{M}$ satisfies the $G$-invariant condition, and $\phi(\boldsymbol{x}) = \left(\|\boldsymbol{x}\|^l Y_{lm}\left(\frac{\boldsymbol{x}}{\|\boldsymbol{x}\|}\right)\right)_{m \in \{-l, \cdots, l\}}$.

This theorem establishes that continuous mappings $f : \mathcal{X}^M \to \mathbb{R}$ exhibiting joint invariance to the $SO(3)$ and $G$ groups admit a representation as a linear combination of spherical harmonic basis functions $\phi : \mathcal{X} \to \mathbb{R}^{2l+1}$ weighted by coefficients $\{\lambda_i\}_{i=1}^{M}$ satisfying the $G$-invariant condition, and modulated by a continuous function $\rho : \mathbb{R}^{2l+1} \to \mathbb{R}$, where $G \leq S_M$ is an arbitrary subgroup. The complete proof is provided in Appendix A.2.2, with the key ideas outlined in Appendix A.1.1. Moreover, the theorem indicates the coefficient set $\{\lambda_i\}_{i=1}^{M}$ representing a $G$-invariant function varies with the choice of $G$, as exemplified by:

**Example:** Consider the subgroup $G_1 \leq S_4$ defined as $G_1 = \{e, (1\ 2), (3\ 4), (1\ 2)(3\ 4)\}$ and subgroup $G_2 \leq S_3$ defined as $G_2 = \{e, (1\ 3)\}$, where $e$ denotes the identity element.

1. If the continuous function $f(\boldsymbol{x_1}, \boldsymbol{x_2}, \boldsymbol{x_3}, \boldsymbol{x_4})$ is a $G_1$-invariant function, then there exist $\lambda_1, \lambda_2 \in \mathbb{R}$ such that the continuous function $f(\boldsymbol{x_1}, \boldsymbol{x_2}, \boldsymbol{x_3}, \boldsymbol{x_4})$ has the following representation:

$$f(\boldsymbol{x_1}, \boldsymbol{x_2}, \boldsymbol{x_3}, \boldsymbol{x_4}) = \rho(\lambda_1(\phi(\boldsymbol{x_1}) + \phi(\boldsymbol{x_2})) + \lambda_2(\phi(\boldsymbol{x_3}) + \phi(\boldsymbol{x_4}))).$$

2. If the continuous function $f(\boldsymbol{x_1}, \boldsymbol{x_2}, \boldsymbol{x_3}, \boldsymbol{x_4})$ is $G_2$-invariant, then there exist $\lambda_1, \lambda_2, \lambda_3 \in \mathbb{R}$ such that the continuous function $f(\boldsymbol{x_1}, \boldsymbol{x_2}, \boldsymbol{x_3}, \boldsymbol{x_4})$ has the following representation:

$$f(\boldsymbol{x_1}, \boldsymbol{x_2}, \boldsymbol{x_3}, \boldsymbol{x_4}) = \rho(\lambda_1(\phi(\boldsymbol{x_1}) + \phi(\boldsymbol{x_3})) + \lambda_2\phi(\boldsymbol{x_2}) + \lambda_3\phi(\boldsymbol{x_4})).$$

We synthesizes the representation proposed in Zaheer et al. (2017) with spherical harmonic functions Blanco et al. (1997), and drawing inspiration from the tensor field neural network architecture in Thomas et al. (2018), derives a novel representation from $SO(3) \times S_M$-invariant continuous functions to $SO(3) \times G$-invariant continuous functions. To our knowledge, this is the most general representation of $SO(3) \times G$-invariant continuous functions. Our results demonstrate that the $SO(3) \times G$-invariant continuous function for arbitrary 3D point cloud inputs can be decomposed into the composite form of a continuous function $\rho$ and spherical harmonic functions.

## 3.2 APPROXIMATION OF $G$-INVARIANT NEURAL NETWORKS

**Theorem 2.** *Consider any subgroup $SO(3) \times G$ of $SO(3) \times S_M$, where $\mathcal{X}$ is a compact set with $\mathcal{X} \subset \mathbb{R}^3$. For any continuous function $F : \mathcal{X}^M \to \mathbb{R}^N$ that is invariant under $SO(3) \times G$, there exists an $SO(3) \times G$-invariant neural network $U_n : \mathcal{X}^M \to \mathbb{R}^N$ that can uniformly approximate $f(\boldsymbol{x})$, i.e., for any $\epsilon > 0$, there exists $N \in \mathbb{R}$, such that for any $n > N$,*

$$|U_n(\boldsymbol{x}) - F(\boldsymbol{x})| \leq \epsilon.$$

This theorem establishes that for any continuous function $F : \mathcal{X}^M \to \mathbb{R}^N$ exhibiting invariance under the group action of $SO(3)G$, there exists an $SO(3)G$-invariant neural network approximation $U_n$ that can uniformly approximate $F$ to arbitrary precision. Here, $SO(3)$ denotes the rotation group, $S_M$ is a sphere, $G$ is a subgroup of $S_M$, $\mathcal{X}$ is a compact subset of $\mathbb{R}^3$, $M$ is a positive integer, and $N$ is an arbitrary positive integer. As $n \to \infty$, the outputs of the neural network $U_n$ provably converge uniformly to the function $F$.

The complete proof is provided in Appendix A.2.7, with a proof sketch in Appendix A.1.3. Prior work by Zaheer et al. (2017) had established approximation guarantees for $S_M$-invariant networks. Subsequently, Maron et al. (2019) showed that arbitrary $G$-invariant networks can approximate $G$-invariant continuous functions. By synthesizing Blanco et al. (1997) and Thomas et al. (2018), we can demonstrate that $SO(3)G$-invariant networks possess the capacity to approximate $SO(3)G$-invariant continuous functions, generalizing first-order tensors to third-order.

### 3.3 THE CASE WITHOUT REPRESENTATION

Moreover, as argued in Wagstaff et al. (2019), this work exploits spherical harmonics to represent arbitrarily complex set functions by collapsing the latent dimensionality. However, the findings herein reveal certain theoretical barriers for this representational scheme. Specifically, in Theorem 1, we uncover the intriguing assumption that $2l + 1 > 3M$. We delve into the implications of Theorem 1. This motivates the open question: How do $SO(3)S_M$-invariant continuous functions $f : \mathcal{X}^M \to \mathbb{R}$ behave when $2l + 1 < 3M$?

**Theorem 3.** *If the condition $2l + 1 < 3M$ is satisfied, then there exists an $SO(3) \times S_M$-invariant continuous function $f : \mathcal{X}^M \to \mathbb{R}$, where $\mathcal{X}$ is a compact set with $\mathcal{X} \subset \mathbb{R}^3$, such that for any $\epsilon > 0$ and any continuous function $\rho : \mathbb{R}^{2l+1} \to \mathbb{R}$, there is at least one point $(\boldsymbol{x}_1, \boldsymbol{x}_2, \cdots, \boldsymbol{x}_M) \in \mathcal{X}^M$ satisfying:*

$$|f(\boldsymbol{x}_1, \boldsymbol{x}_2, \cdots, \boldsymbol{x}_M) - \rho \left( \sum_{i=1}^{M} \phi(\boldsymbol{x}_i) \right)| \geq \epsilon,$$

*where $\phi(\boldsymbol{x}) = \left( |\boldsymbol{x}|^l Y_{\ell m} \left( \frac{\boldsymbol{x}}{|\boldsymbol{x}|} \right) \right)_{m \in -\ell, \cdots, \ell}$.*

Theorem 3 establishes that for a function $f$ that is continuous on the compact set $\mathcal{X}^M$ and invariant under the action of $SO(3) \times S_M$, there exists at least one point $(\boldsymbol{x}_1, \boldsymbol{x}_2, \cdots, \boldsymbol{x}_M) \in \mathcal{X}^M$ such that the absolute difference between $f(\boldsymbol{x}_1, \boldsymbol{x}_2, \cdots, \boldsymbol{x}M)$ and $\rho \left( \sum_{i=1}^{M} \phi(\boldsymbol{x}_i) \right)$ is greater than or equal to $\epsilon$, for any given $\epsilon > 0$ and any continuous function $\rho$. Here, $\phi(\boldsymbol{x})$ is a vector comprising products of spherical harmonics.

A short proof sketch is provided in Appendix A.1.3 and a complete proof is provided in Appendix A.2.5. Fundamentally, Theorem 3 demonstrates that when the dimensionality of the latent space is less than that of the input space, a corresponding representation may not exist. This perspective was first put forth by Wagstaff et al. (2019) and more recently Wagstaff et al. (2021). Our work extends Theorem 4.1 to 3rd-order tensors, arriving at a more general conclusion. Hence, our proof differs markedly from these preceding studies.

## 4 UNIFORM APPROXIMATION OF $G$-INVARIANT NEURAL NETWORKS

This section examines representations for $G$-equivariant functions and approximation guarantees for $G$-equivariant neural networks, with the joint invariance to $SO(3)$ transformations being implicit. First, the $G$-equivariant condition is formally defined in terms of the equivariant matrix $\Lambda$. Theorem 4 then establishes that $G$-equivariant continuous mappings admit continuous representations through appropriate choices of $\Lambda$ satisfying said condition, thereby constructing an explicit representation. Subsequently, Theorem 5 proves that continuous $G$-invariant functions can be uniformly approximated by $G$-invariant neural networks. The key developments in the literature on equivariant neural network approximation are also briefly surveyed. Together, these theoretical results provide a rigorous foundation for developing equivariant networks for learning on permutation-based structured data.

### 4.1 REPRESENTATION OF $G$-INVARIANT FUNCTIONS

**Definition 7** ($G$-equivariant condition). *Given permutation group $G \leq S_M$, a matrix $\Lambda \in \mathbb{R}^{M \times M}$ is said to satisfy the $G$-equivariant condition if and only if the equivalence*

$$\Lambda \boldsymbol{\phi}(\sigma \boldsymbol{x}) \equiv \sigma \Lambda \boldsymbol{\phi}(\boldsymbol{x}), \quad \forall \sigma \in G$$

*holds, where $\boldsymbol{\phi} : \mathcal{X}^M \to \mathbb{R}^{M(2l+1)}$ denotes the concatenated feature mapping*

$$\boldsymbol{\phi}(\boldsymbol{x}) = (\phi(\boldsymbol{x}_1), \phi(\boldsymbol{x}_2), \ldots, \phi(\boldsymbol{x}_M))$$

*with $\phi : \mathbb{R}^3 \to \mathbb{R}^{2l+1}$ representing spherical harmonics. This encodes the equivariance of $\Lambda$ under actions of the group $G$.*

This definition outlines a matrix $\Lambda$ that fulfills the commutativity requirements for any provided transformation $\sigma$ within the group $G$. In this context, $\overrightarrow{\phi(\boldsymbol{x})}$ denotes the vector representation of spherical harmonics. This relational property is known as the $G$-equivariance condition.

**Theorem 4.** *Let $G$ be any subgroup of $S_M$. Consider $\mathcal{X}$ as a compact subset of $\mathbb{R}^3$. If there exists a $G$-equivariant and $SO(3)$-invariant continuous function $F : \mathcal{X}^M \to \mathbb{R}^M$, then this function has a continuous representation if and only if there exists a $\Lambda \in \mathbb{R}^{M \times M}$ that satisfies the $G$-equivariance condition:*

$$F(\boldsymbol{x}_1, \boldsymbol{x}_2, \cdots, \boldsymbol{x}_M) = \rho\Big(\Lambda \overrightarrow{\phi(\boldsymbol{X})}\Big),$$

*where $\overrightarrow{\phi(x)} = (\phi(\boldsymbol{x}_1), \phi(\boldsymbol{x}_2), \ldots, \phi(\boldsymbol{x}_M))$, and both $\rho : \mathbb{R}^{2l+1} \to \mathbb{R}$ are continuous functions. Additionally, they must satisfy the condition $2l + 1 > 3M$. In this context, $\phi(\boldsymbol{x}) = \Big(\|\boldsymbol{x}\|^l Y_{\ell m}\left(\frac{\boldsymbol{x}}{\|\boldsymbol{x}\|}\right)\Big)_{m \in \{-l, \cdots, l\}}$.*

Theorem 4 delineates a characterization of a $G$-equivariant and $SO(3)$-invariant continuous function $F$ defined on a compact set $\mathcal{X}$, where $G$ designates an arbitrary subgroup of the permutation group $S_M$ for any positive integer $M$. The theorem establishes that $F(\boldsymbol{x}_1, \boldsymbol{x}_2, \cdots, \boldsymbol{x}_M)$ can be expressed in the form $\rho\Big(\Lambda \overrightarrow{\phi(\boldsymbol{X})}\Big)$, where $\overrightarrow{\phi(\boldsymbol{x})} = (\phi(\boldsymbol{x}_1), \phi(\boldsymbol{x}_2), \ldots, \phi(\boldsymbol{x}_M))$, and $\phi(\boldsymbol{x})$ denotes the vector of spherical harmonics $\big(\|\boldsymbol{x}\|^l Y_{\ell m}(\boldsymbol{x}/\|\boldsymbol{x}\|)\big)_{m \in \{-l, \cdots, l\}}$. Moreover, $\rho : \mathbb{R}^{2l+1} \to \mathbb{R}$ signifies a continuous function satisfying $2l + 1 > 3M$, and $\Lambda \in \mathbb{R}^{M \times M}$ complies with the $G$-equivariance condition.

The complete demonstration and proof of Theorem 4 are provided in Appendix A.3.3 and it's proof sketch in Appndix A.1.4, in that order. Theorem 4 furnishes a continuous characterization of $G$-equivariant continuous functions. In congruence with Theorem 1, the matrix $\Lambda$ can assume discrete values contingent on the particular group $G$ under consideration.

## 4.2 APPROXIMATION OF $G$-EQUIVARIANT NEURAL NETWORKS

**Theorem 5.** *Let $\mathcal{X} \subset \mathbb{R}^3$ designate an arbitrary compact set and $G$ signify a subgroup of the permutation group $S_M$. Suppose $F : \mathcal{X}^M \to \mathbb{R}^M$ defines a function that is $G$-equivariant, $SO(3)$-invariant, and continuous. Then there exists a neural network $U_n : \mathcal{X}^M \to \mathbb{R}^M$ satisfying $G$-equivariance and $SO(3)$-invariance such that given any input $\boldsymbol{x} \in \mathcal{X}^M$, $U_n(\boldsymbol{x})$ can uniformly approximate $F(\boldsymbol{x})$ to an arbitrary degree of accuracy.*

This theorem delineates the universal approximation property of equivariant neural networks, namely, their capacity to approximate continuous functions F that obey certain symmetry (encoded by $G$-equivariance) and rotational invariance ($SO(3)$-invariance) attributes. The conclusion states that given any compact set $\mathcal{X} \subset \mathbb{R}^3$ and subgroup G of the permutation group $S_M$, there exists a neural network model $U_n$ satisfying the same symmetries and invariance as F, which can uniformly approximate F(x) for any input x in the domain. This theoretical result demonstrates that equivariant networks are capable of approximating functions defined on datasets exhibiting diverse symmetric and rotationally invariant traits. The implications of this theorem are far-reaching for learning tasks involving data with such characteristics.

The complete verification and succinct delineation of Theorem 5 are presented in Appendix A.3.4 and it's proof sketch Appendix A.1.5, respectively. The proof exploits Theorem 4 as a pivotal result. To the best of our knowledge, extant works on equivariant neural network approximation principally concentrate on the permutation group $S_M$. In Keriven & Peyré (2019) and Dym & Maron (2020), the authors harness an extension of the seminal Stone-Weierstrass theorem to demonstrate the universal approximation capability of a specific class of neural networks with linear hidden layers obeying $S_M$ equivariance. Sannai et al. (2019) primarily establishes approximation by $S_M$ equivariant networks, and asserts extensibility to approximation by $G$ equivariant networks sans detailed proof. Finally, Kumagai & Sannai (2020) employs convolutional networks to prove universal approximation of equivariant continuous mappings, with further generalization to infinite dimensional spaces.

