REFERENCES

Milton Abramowitz, Irene A Stegun, and Robert H Romer. Handbook of mathematical functions with formulas, graphs, and mathematical tables, 1988.

Andrew R Barron. Approximation and estimation bounds for artificial neural networks. *Machine Learning*, 14(1):115–133, 1994.

Miguel A Blanco, Manuel Flórez, and Margarita Bermejo. Evaluation of the rotation matrices in the basis of real spherical harmonics. *Journal of Molecular Structure: Theochem*, 419(1-3):19–27, 1997.

Alexander Bogatskiy, Brandon Anderson, Jan Offermann, Marwah Roussi, David Miller, and Risi Kondor. Lorentz group equivariant neural network for particle physics. In *International Conference on Machine Learning*, pp. 992–1002. PMLR, 2020.

Georg Bökman, Fredrik Kahl, and Axel Flinth. Zz-net: A universal rotation equivariant architecture for 2d point clouds. In *Proceedings of the IEEE/CVF Conference on Computer Vision and Pattern Recognition*, pp. 10976–10985, 2022.

Ashesh Chattopadhyay, Mustafa Mustafa, Pedram Hassanzadeh, and Karthik Kashinath. Deep spatial transformers for autoregressive data-driven forecasting of geophysical turbulence. In *Proceedings of the 10th International Conference on Climate Informatics*, pp. 106–112, 2020.

Gregory S Chirikjian. *Engineering applications of noncommutative harmonic analysis: with emphasis on rotation and motion groups*. CRC press, 2000.

Taco Cohen and Max Welling. Group equivariant convolutional networks. In *International Conference on Machine Learning*, pp. 2990–2999. PMLR, 2016.

George Cybenko. Approximation by superpositions of a sigmoidal function. *Mathematics of Control, Signals and Systems*, 2(4):303–314, 1989.

Ingrid Daubechies, Ronald DeVore, Simon Foucart, Boris Hanin, and Guergana Petrova. Nonlinear approximation and (deep) relu networks. *Constructive Approximation*, pp. 1–46, 2021.

Ronald DeVore, Boris Hanin, and Guergana Petrova. Neural network approximation. *Acta Numerica*, 30:327–444, 2021.

David Steven Dummit and Richard M Foote. *Abstract algebra*, volume 3. Wiley Hoboken, 2004.

Nadav Dym and Haggai Maron. On the universality of rotation equivariant point cloud networks. *arXiv preprint arXiv:2010.02449*, 2020.

Carlos Esteves. Theoretical aspects of group equivariant neural networks. *arXiv preprint arXiv:2004.05154*, 2020.

Carlos Esteves, Christine Allen-Blanchette, Ameesh Makadia, and Kostas Daniilidis. Learning so (3) equivariant representations with spherical cnns. In *Proceedings of the European Conference on Computer Vision (ECCV)*, pp. 52–68, 2018.

Ben Finkelshtein, Chaim Baskin, Haggai Maron, and Nadav Dym. A simple and universal rotation equivariant point-cloud network. In *Topological, Algebraic and Geometric Learning Workshops 2022*, pp. 107–115. PMLR, 2022.

Fabian Fuchs, Daniel Worrall, Volker Fischer, and Max Welling. Se (3)-transformers: 3d roto-translation equivariant attention networks. *Advances in Neural Information Processing Systems*, 33:1970–1981, 2020.

Ken-Ichi Funahashi. On the approximate realization of continuous mappings by neural networks. *Neural Networks*, 2(3):183–192, 1989.

Ken-ichi Funahashi and Yuichi Nakamura. Approximation of dynamical systems by continuous time recurrent neural networks. *Neural Networks*, 6(6):801–806, 1993.

Liyao Gao, Hongshan Li, Zheying Lu, and Guang Lin. Rotation-equivariant convolutional neural network ensembles in image processing. In *Adjunct Proceedings of the 2019 ACM International Joint Conference on Pervasive and Ubiquitous Computing and Proceedings of the 2019 ACM International Symposium on Wearable Computers*, pp. 551–557, 2019.

Yulan Guo, Hanyun Wang, Qingyong Hu, Hao Liu, Li Liu, and Mohammed Bennamoun. Deep learning for 3d point clouds: A survey. *IEEE Transactions on Pattern Analysis and Machine Intelligence*, 43(12):4338–4364, 2020.

Kurt Hornik. Some new results on neural network approximation. *Neural Networks*, 6(8):1069–1072, 1993.

Kurt Hornik, Maxwell Stinchcombe, and Halbert White. Multilayer feedforward networks are universal approximators. *Neural Networks*, 2(5):359–366, 1989.

Riley Jackson, Wenyuan Zhang, and Jason Pearson. Tsnet: predicting transition state structures with tensor field networks and transfer learning. *Chemical Science*, 12(29):10022–10040, 2021.

Nicolas Keriven and Gabriel Peyré. Universal invariant and equivariant graph neural networks. *Advances in Neural Information Processing Systems*, 32:7092–7101, 2019.

Anthony W Knapp. Representation theory of semisimple groups: an overview based on examples. 2001.

Wataru Kumagai and Akiyoshi Sannai. Universal approximation theorem for equivariant maps by group cnns. *arXiv preprint arXiv:2012.13882*, 2020.

Hannah Lawrence. Barron's theorem for equivariant networks. In *NeurIPS 2022 Workshop on Symmetry and Geometry in Neural Representations*, 2022.

Moshe Leshno, Vladimir Ya Lin, Allan Pinkus, and Shimon Schocken. Multilayer feedforward networks with a nonpolynomial activation function can approximate any function. *Neural networks*, 6(6):861–867, 1993.

Qianxiao Li, Ting Lin, and Zuowei Shen. Deep neural network approximation of invariant functions through dynamical systems. *arXiv preprint arXiv:2208.08707*, 2022.

Ziyao Li, Shuwen Yang, Guojie Song, and Lingsheng Cai. Hamnet: conformation-guided molecular representation with hamiltonian neural networks. *arXiv preprint arXiv:2105.03688*, 2021.

Tianyu Ma, Ajay Gupta, and Mert R Sabuncu. Volumetric landmark detection with a multi-scale shift equivariant neural network. In *2020 IEEE 17th International Symposium on Biomedical Imaging (ISBI)*, pp. 981–985. IEEE, 2020.

Haggai Maron, Ethan Fetaya, Nimrod Segol, and Yaron Lipman. On the universality of invariant networks. In *International Conference on Machine Learning*, pp. 4363–4371. PMLR, 2019.

Haggai Maron, Or Litany, Gal Chechik, and Ethan Fetaya. On learning sets of symmetric elements. In *International Conference on Machine Learning*, pp. 6734–6744. PMLR, 2020.

Adrien Poulenard and Leonidas J Guibas. A functional approach to rotation equivariant non-linearities for tensor field networks. In *Proceedings of the IEEE/CVF Conference on Computer Vision and Pattern Recognition*, pp. 13174–13183, 2021.

Adrien Poulenard, Marie-Julie Rakotosaona, Yann Ponty, and Maks Ovsjanikov. Effective rotation-invariant point cnn with spherical harmonics kernels. In *2019 International Conference on 3D Vision (3DV)*, pp. 47–56. IEEE, 2019.

Charles R Qi, Hao Su, Kaichun Mo, and Leonidas J Guibas. Pointnet: Deep learning on point sets for 3d classification and segmentation. In *Proceedings of the IEEE Conference on Computer Vision and Pattern Recognition*, pp. 652–660, 2017.

CR Qi, L Yi, H PointNet+ Su, and L PointNet+ Guibas. Deep hierarchical feature learning on point sets in a metric space. arxiv 2017. *arXiv preprint arXiv:1706.02413*.

Akiyoshi Sannai, Yuuki Takai, and Matthieu Cordonnier. Universal approximations of permutation invariant/equivariant functions by deep neural networks. *arXiv preprint arXiv:1903.01939*, 2019.

Nimrod Segol and Yaron Lipman. On universal equivariant set networks. *arXiv preprint arXiv:1910.02421*, 2019.

Sho Sonoda and Noboru Murata. Neural network with unbounded activation functions is universal approximator. *Applied and Computational Harmonic Analysis*, 43(2):233–268, 2017.

Nathaniel Thomas, Tess Smidt, Steven Kearnes, Lusann Yang, Li Li, Kai Kohlhoff, and Patrick Riley. Tensor field networks: Rotation-and translation-equivariant neural networks for 3d point clouds. *arXiv preprint arXiv:1802.08219*, 2018.

Oriol Vinyals, Samy Bengio, and Manjunath Kudlur. Order matters: Sequence to sequence for sets. *arXiv preprint arXiv:1511.06391*, 2015.

Edward Wagstaff, Fabian Fuchs, Martin Engelcke, Ingmar Posner, and Michael A Osborne. On the limitations of representing functions on sets. In *International Conference on Machine Learning*, pp. 6487–6494. PMLR, 2019.

Edward Wagstaff, Fabian B Fuchs, Martin Engelcke, Michael A Osborne, and Ingmar Posner. Universal approximation of functions on sets. *arXiv preprint arXiv:2107.01959*, 2021.

Daniel E Worrall, Stephan J Garbin, Daniyar Turmukhambetov, and Gabriel J Brostow. Harmonic networks: Deep translation and rotation equivariance. In *Proceedings of the IEEE Conference on Computer Vision and Pattern Recognition*, pp. 5028–5037, 2017.

Dmitry Yarotsky. Universal approximations of invariant maps by neural networks. *Constructive Approximation*, pp. 1–68, 2021.

Dmitry Yarotsky. Universal approximations of invariant maps by neural networks. *Constructive Approximation*, 55(1):407–474, 2022.

Manzil Zaheer, Satwik Kottur, Siamak Ravanbakhsh, Barnabas Poczos, Ruslan Salakhutdinov, and Alexander Smola. Deep sets. *arXiv preprint arXiv:1703.06114*, 2017.

Zhiyuan Zhang, Binh-Son Hua, David W Rosen, and Sai-Kit Yeung. Rotation invariant convolutions for 3d point clouds deep learning. In *2019 International Conference on 3D Vision (3DV)*, pp. 204–213. IEEE, 2019.

## A APPENDIX

### A.1 PROOF SKETCH

#### A.1.1 PROOF SKETCH OF THEOREM 1

We now provide a proof sketch for Theorem 1. We first consider the action of the permutation group, which can be divided into two steps. First, similar to the conclusion in Zaheer et al. (2017), we examine the simple symmetric group $S_M$, from which a theorem involving composite functions of spherical harmonics and continuous functions can be obtained. The complete proof is deferred to Appendix A.2.3.

For the second step, by Lemma 1, any $\sigma \in G \subset S_M$ can be decomposed into a product of disjoint cycles. Hence, it suffices to show that any rotation can be expressed as permutations. The complete details are provided in Appendix A.2.2.

**Proof Stage 1: The symmetric group $S_M$ can be continuously represented by spherical harmonics.**

**Theorem 6.** *Consider $\mathcal{X}$ as a compact set with $\mathcal{X} \subset \mathbb{R}^3$. A continuous function $f : \mathcal{X}^M \to \mathbb{R}$ is $SO(3) \times S_M$-invariant if and only if $f$ admits the representation:*

$$f(\boldsymbol{x}_1, \boldsymbol{x}_2, \cdots, \boldsymbol{x}M) = \rho\left(\sum_{i=1}^{M} \phi(\boldsymbol{x}_i)\right),$$

*where $\rho : \mathbb{R}^{2l+1} \to \mathbb{R}$ is a continuous function, $2l+1 > 3M$, and*

$$\phi(\boldsymbol{x}) = \left(|\boldsymbol{x}|^l Y_{\ell m}\left(\frac{\boldsymbol{x}}{|\boldsymbol{x}|}\right)\right)_{m \in -\ell, \cdots, \ell}.$$

The complete proof is relegated to Appendix A.2.3. Through an integrated framework synthesizing key elements of Zaheer et al. (2017), Blanco et al. (1997), and Thomas et al. (2018), we establish a hitherto unknown representation theorem for continuous functions exhibiting invariance under the action of the symmetric group $S_M$. Furthermore, we adopt the nomenclature of designating $\sum_{i=1}^{M} \phi(\boldsymbol{x}_i)$ as constituting the **latent space**.

Our proof of $SO(3)$-invariance critically builds upon and extends Theorem 2 from the seminal work of Zaheer et al. (2017). By scrutinizing the permutation invariance properties of $\rho\left(\sum_{i=1}^{M} \phi(\boldsymbol{x}_i)\right)$, we discern that the sufficient condition of Theorem 6 is immediately satisfied if $f(\boldsymbol{x}_1, \boldsymbol{x}_2, \cdots, \boldsymbol{x}_M) = \rho\left(\sum_{i=1}^{M} \phi(\boldsymbol{x}_i)\right)$. Thus, the brunt of the technical effort resides in establishing necessity. Our overarching proof strategy is by contradiction - we aim to demonstrate that in the absence of an equivariant relationship between the points $(\boldsymbol{x}_1^1, \boldsymbol{x}_2^1, \ldots, \boldsymbol{x}_M^1)$ and $(\boldsymbol{x}_1^2, \boldsymbol{x}_2^2, \ldots, \boldsymbol{x}_M^2)$, the theorem statement uniquely characterizes the permutation invariant continuous functions. i.e,

$$\sum_{i=1}^{M} \phi(\boldsymbol{x}_i^1) \neq \sum_{i=1}^{M} \phi(\boldsymbol{x}_i^2).$$

That is, we demonstrate that $\Phi(\boldsymbol{x}) = \sum_{i=1}^{M} \phi(\boldsymbol{x}_i)$ constitutes an injective mapping. Finally, an appropriate continuous function $\rho$ can be selected to conclude the proof. $SO(3)$-invariance is guaranteed in Dym & Maron (2020). Theorem 6 establishes that $SO(3)S_M$-invariant continuous functions $f : \mathcal{X}^M \to \mathbb{R}$ can be represented through compositions of continuous functions and spherical harmonics.

Furthermore, we establish that the representation generalizes to vector-valued functions $F : \mathcal{X}^M \to \mathbb{R}^N$, as elucidated in Corollary 2 contained within Appendix A.2.4. The proof leverages a decompo-

sition of $F = (f_1, f_2, \ldots, f_N)$ into constituent component functions $f_i : \mathcal{X}^M \to \mathbb{R}$ for $i \in [1, N]$, each of which satisfies Theorem 6. The modular structure thereby facilitates a direct generalization to higher dimensions.

**Proof Stage 2: The general group can be continuously represented by spherical harmonics through induction.**

Pertaining to establishing $G$-invariance, we shall elucidate that $\forall \sigma \in G$, an appropriate continuous representation exists. We approach this in a modular fashion by examining permutations; as the base case, we analyze the $n = 2$ permutation corresponding to a simple transposition or exchange.Pertaining to establishing $G$-invariance, we shall elucidate that $\forall \sigma \in G$, an appropriate continuous representation exists. We approach this in a modular fashion by examining permutations; as the base case, we analyze the $n = 2$ permutation corresponding to a swap.

The elemental transposition $(i, j)$ induces a group $G_{(i,j)} = (i, j), e$ under function composition. It follows that $G_{(i,j)}$-invariant continuous functions reduce to investigating $(i, j)$-invariant continuous functions, thereby circumscribing our analysis to permutations of the variables $x_i$ and $x_j$. Leveraging the group structure of $G_{(i,j)} = (i, j), e = S_2$ generated by the $n = 2$ transposition, we derive the ensuing continuous representation theorem for $S_2$-invariant continuous functions:

$$f(\boldsymbol{x}_1, \boldsymbol{x}_2, \cdots, \boldsymbol{x}_M) = \rho\Big(\sum_{s=1, s \neq i, j}^{M} \lambda_s \phi(\boldsymbol{x}_s) + \lambda(\phi(\boldsymbol{x}_i) + \phi(\boldsymbol{x}_j))\Big).$$

Further details can be found in Corollary 1 of Appendix A.2.2. This corollary delineates that $\lambda_s{}_{s=1, s \neq i, j}^{M}$ and $\lambda$ satisfy the $G(i, j)$-invariance conditions.

Assume we have established a continuous representation for the permutation $(i_1, i_2, \ldots, i_{n-1})$. We now consider the inductive case of the $n$-permutation $(i_1, i_2, \ldots, i_n)$. Appealing to Lemma 1 (proof in Appendix A.2.6), we discern that the $n$-permutation admits the decomposition $(i_1, i_2, \ldots, i_n) = (i_1, i_n)(i_1, i_2, \ldots, i_{n-1})$. By recursively applying this scheme, it becomes evident that any $n$-permutation can be expressed as the composition of an $(n-1)$-permutation and a transposition. Specifically, $(i_1, i_2, \ldots, i_n) = (i_1, i_2)(i_1, i_3, i_4, \ldots, i_n)$, where $(i_1, i_3, i_4, \ldots, i_{n-1})$ constitutes an $(n-1)$-permutation. This inductive reasoning establishes that $n$-permutations possess continuous representations.

Lemma 1 establishes that an arbitrary element $\sigma \in G$ admits a representation as a composition of disjoint permutations. Therefore, the continuous representability of $n$-permutations immediately propagates to general $\sigma \in G$. We defer the technical apparatus pertaining to constructing $\rho$ to guarantee the requisite $SO(2l+1)$-invariance to Dym & Maron (2020). It thus follows that $SO(3) \times G$-invariant continuous functions are characterized by the representation:

$$f(\boldsymbol{x}_1, \boldsymbol{x}_2, \cdots, \boldsymbol{x}_M) = \rho\left(\sum_{i=1}^{M} \lambda_i \phi(\boldsymbol{x}_i)\right),$$

### A.1.2 PROOF SKETCH OF THEOREM 2

In this proof, we first leverage Theorem 1 to establish the representation form of equivariant continuous functions and extend it to equivariant neural networks. We introduce continuous functions $\rho$ and $\rho_n$, both defined on $\mathbb{R}^{2l+1}$ with $2l + 1 > 3M$. The functions $\lambda_i$ are chosen to satisfy the $G$-condition, and we define $\phi(\boldsymbol{x})$ as a function involving spherical harmonics and the Euclidean norm of $\boldsymbol{x}$. The core concept lies in $\Phi(\boldsymbol{x})$, which is the sum of weighted $\phi(\boldsymbol{x}_i)$ terms, where $\boldsymbol{x}_i$ are data points from the dataset $\mathcal{X}^M$. The continuity of $\phi$ ensures that $\Phi(\mathcal{X}^M)$ is compact.

In the second step, we aim to approximate the continuous functions $\rho$ and $\rho_n$ using neural networks. We invoke the universal approximation theorem for neural networks, which guarantees the existence of a sequence $\rho_n$ that approximates $\rho$. Crucially, the compactness of $\mathcal{X}^M$ plays a pivotal role here, as the continuity of $\Phi$ implies that $\Phi(\mathcal{X}^M)$ is also compact. This property allows us to apply the Stone-Weierstrass theorem, concluding that there exists a sequence $\rho_n$ that uniformly approximates $\rho$ on the compact set $\Phi(\mathcal{X}^M)$. This two-step approach demonstrates how the representation of

equivariant functions and the use of compact sets enable us to establish the desired result regarding the approximation of continuous functions by neural networks.

### A.1.3 PROOF SKETCH OF THEOREM 3

**Proof Stage 1: The Non-injectivity of** $\Phi(\boldsymbol{x}) = \sum\limits_{i=1}^{M} \phi(\boldsymbol{x}_i)$ **as a Mapping from** $\mathcal{X}^M$ **to** $\mathbb{R}^{2l+1}$ **Under the Regime** $3M > 2l + 1$.

Through a proof by contradiction, we establish that the mapping $\Phi(\boldsymbol{x}) = \sum\limits_{i=1}^{M} \phi(\boldsymbol{x}_i)$ cannot be injective. The crux of the argument is that if $\Phi$ were injective, by virtue of $\phi(\boldsymbol{x})$ being continuous, $\Phi$ would constitute an injective continuous function from the higher dimensional space $\mathcal{X}^M$ to the lower dimensional space $\mathbb{R}^{2l+1}$. However, this is precluded by the topological obstruction that continuous injective mappings cannot exist from higher to lower dimensions.

We shall construct a contradiction by exhibiting two distinct points $(\boldsymbol{x}_1^1, \boldsymbol{x}_2^1, \cdots, \boldsymbol{x}_M^1) \neq (\boldsymbol{x}_1^2, \boldsymbol{x}_2^2, \cdots, \boldsymbol{x}_M^2) \in \mathcal{X}^M$ that satisfy:

$$\sum_{i=1}^{M} \phi(\boldsymbol{x}_i^1) = \sum_{i=1}^{M} \phi(\boldsymbol{x}_i^2).$$

**Proof Stage 2: The Existence of Discrepant Points** $(\boldsymbol{x}_1, \boldsymbol{x}_2, \cdots, \boldsymbol{x}_M) \in \mathcal{X}^M$ **Satisfying the Lower Bound for Arbitrary Functions** $f$

$$\left| f(\boldsymbol{x}_1, \boldsymbol{x}_2, \cdots, \boldsymbol{x}_M) - \rho\left(\sum_{i=1}^{M} \phi(\boldsymbol{x}_i)\right) \right| \geq \epsilon.$$

Given $\|(\boldsymbol{x}_1^1, \boldsymbol{x}_2^1, \cdots, \boldsymbol{x}_M^1) \neq (\boldsymbol{x}_1^2, \boldsymbol{x}_2^2, \cdots, \boldsymbol{x}_M^2)\|$ along with the collision $\sum\limits_{i=1}^{M} \phi(\boldsymbol{x}_i^1) = \sum\limits_{i=1}^{M} \phi(\boldsymbol{x}_i^2)$ whose existence was established via the non-injectivity proof in Stage 1, we exhibit the discrepancy by constructing:

$$f(\boldsymbol{x}_1, \boldsymbol{x}_2, \cdots, \boldsymbol{x}_M) = y_1 \frac{\|(\boldsymbol{x}_1, \boldsymbol{x}_2, \cdots, \boldsymbol{x}_M) - (\boldsymbol{x}_1^2, \boldsymbol{x}_2^2, \cdots, \boldsymbol{x}_M^2)\|}{\|(\boldsymbol{x}_1^1, \boldsymbol{x}_2^1, \cdots, \boldsymbol{x}_M^1) - (\boldsymbol{x}_1^2, \boldsymbol{x}_2^2, \cdots, \boldsymbol{x}_M^2)\|}$$
$$+ y_2 \frac{\|(\boldsymbol{x}_1, \boldsymbol{x}_2, \cdots, \boldsymbol{x}_M) - (\boldsymbol{x}_1^1, \boldsymbol{x}_2^1, \cdots, \boldsymbol{x}_M^1)\|}{\|(\boldsymbol{x}_1^1, \boldsymbol{x}_2^1, \cdots, \boldsymbol{x}_M^1) - (\boldsymbol{x}_1^2, \boldsymbol{x}_2^2, \cdots, \boldsymbol{x}_M^2)\|}$$

By construction, $f$ maps the points $(\boldsymbol{x}_1^1, \boldsymbol{x}_2^1, \cdots, \boldsymbol{x}_M^1)$ and $(\boldsymbol{x}_1^2, \boldsymbol{x}_2^2, \cdots, \boldsymbol{x}_M^2)$ to disparate values $y_1$ and $y_2$ respectively, where $y_1 \neq y_2$. It therefore follows that any putative continuous function $\rho$ is fundamentally unable to equalize these outputs, i.e.,

$$\left| f(\boldsymbol{x}_1, \boldsymbol{x}_2, \cdots, \boldsymbol{x}_M) - \rho\left(\sum_{i=1}^{M} \phi(\boldsymbol{x}_i)\right) \right| \geq \epsilon.$$

### A.1.4 PROOF SKETCH OF THEOREM 4

I shall provide an outline of the proof sketch for Theorem 4:

Firstly, we analyze the action of the permutation group. Regarding the permutation group, the proof is bifurcated into two steps. In the initial step, analogous to the conclusion in [1], we first examine the simple symmetric group $S_M$ and derive a theorem connecting the composition of spherical harmonics and a continuous function by relating it to functions invariant under $\text{Stab}_{S_M}(k)$. The full proof is furnished in Appendix A.3.1.

In the second step, for any $\sigma \in G \subset S_M$, according to Lemma 1, $\sigma$ can be decomposed into a product of disjoint rotation. Therefore, we only need to validate that any rotation can be represented by permutations. The complete verification is provided in Appendix A.3.3.

**Proof Stage 1: Representation of $S_M$-Equivariant and $SO(3)$-Invariant Continuous Functions via a Latent Space.**

**Theorem 7.** *For any compact set $\mathcal{X} \subset \mathbb{R}^3$, a continuous function $F : \mathcal{X}^M \to \mathbb{R}^M$ is $S_M$-equivariant and $SO(3)$-invariant if and only if there exists a matrix $\Lambda$ such that $F : \mathcal{X}^M \to \mathbb{R}^M$ admits a continuous representation*

$$F(\boldsymbol{x}_1, \boldsymbol{x}_2, \cdots, \boldsymbol{x}_M) = \rho\left(\Lambda \overrightarrow{\phi(\boldsymbol{x})}\right),$$

*where $\overrightarrow{\phi(\boldsymbol{x})} = (\phi(\boldsymbol{x}_1), \phi(\boldsymbol{x}_2), \dots, \phi(\boldsymbol{x}_M))$, $\rho : \mathbb{R}^{2l+1} \to \mathbb{R}^M$ designates a continuous function, $2l + 1 > 3M$,*

$$\Lambda = \begin{bmatrix} \lambda_1 & \lambda_2 & \cdots & \lambda_2 \\ \lambda_2 & \lambda_1 & \cdots & \lambda_2 \\ \vdots & \vdots & \ddots & \vdots \\ \lambda_2 & \lambda_2 & \cdots & \lambda_1 \end{bmatrix},$$

*and $\phi(\boldsymbol{x}) = \left(\|\boldsymbol{x}\|^l Y_{\ell m}\left(\boldsymbol{x}/\|\boldsymbol{x}\|\right)\right)_{m \in \{-l, \cdots, l\}}$.*

The complete verification is furnished in Appendix A.3.1, wherein we harness spherical harmonics and continuous characterizations to generalize one-dimensional continuous representations to 3D tensors. This result enhances Proposition 3.1 present in Sannai et al. (2019).

Here is an outline of the sufficiency proof: First consider the permutation group. For any $\sigma \in S_M$, we can verify and calculate:

$$F(\sigma \boldsymbol{x}) = \rho\left(\Lambda \overrightarrow{\phi(\sigma \boldsymbol{x})}\right) = \rho\left(\sigma \Lambda \overrightarrow{\phi(\boldsymbol{x})}\right) = \sigma \rho\left(\Lambda \overrightarrow{\phi(\boldsymbol{x})}\right)$$

Therefore, $F(\boldsymbol{x}) = \rho\left(\Lambda \overrightarrow{\phi(\boldsymbol{x})}\right)$ is $S_M$-equivariant. $SO(3)$-invariance is guaranteed by Dym & Maron (2020).

We establish the necessity of Theorem 7 by introducing the definition of $\text{Stab}_{S_M}(k)$. Here, $\text{Stab}_{S_M}(k)$ signifies the subgroup fixing coordinate $k \in \{1, 2, \dots, M\}$, comprising elements $s \in S_M$ satisfying $s(k) = k$. Denoting $F(\boldsymbol{x}) = (f_1(\boldsymbol{x}), f_2(\boldsymbol{x}), \dots, f_M(\boldsymbol{x}))$ and $\sigma = (i\ j) \in S_M$, by verifying $j$ one by one, we can demonstrate that $f_j(\boldsymbol{x})$ defines a $SO(3) \times \text{Stab}_{S_M}(i)$-invariant function. Consequently, $S_M$-equivariant and $SO(3)$-invariant functions can be transformed into the form of $SO(3) \times \text{Stab}_{S_M}(k)$-invariant functions. More details on the characterization of $SO(3) \times \text{Stab}_{S_M}(k)$-invariant functions are provided in Corollary 3 of Appendix A.3.1. Here we only present the conclusion regarding

$$F(\boldsymbol{x}_1, \boldsymbol{x}_2, \cdots, \boldsymbol{x}_M) = \rho\left(\Lambda \overrightarrow{\phi(\boldsymbol{x})}\right),$$

and

$$\Lambda = \begin{bmatrix} \lambda_1 & \lambda_2 & \cdots & \lambda_2 \\ \lambda_2 & \lambda_1 & \cdots & \lambda_2 \\ \vdots & \vdots & \ddots & \vdots \\ \lambda_2 & \lambda_2 & \cdots & \lambda_1 \end{bmatrix}.$$

**Proof Stage 2: Inductive Demonstration that $G$-Equivariant Functions Admit a Continuous Representation.**

The sufficiency is established via direct substitution verification. To prove necessity, we use induction on the size of cycles n.

Base case $(n = 2)$: Show swaps $(i\ j)$ form a group $G_{(ij)}$. By previous results, we have a continuous representation for $G_{(i\ j)}$-equivariant functions.

Inductive step: Assume we have a representation for $(n-1)$-cycles. Use Lemma to decompose an n-cycle into an $(n-1)$-cycle and a swap. By inductive hypothesis and base case, we can characterize $n$-cycle equivariant functions.

Therefore, by induction, we can characterize $G$-equivariant continuous functions for any $G$ made up of cycles. The key is decomposing into smaller cycles and using the continuous representation for those simpler groups.

### A.1.5 Proof Sketch of Theorem 5

The proof can be bifurcated into two steps. First, we show that a continuously differentiable equivariant function can be represented in the form given by Theorem 4, denoted as $F(\mathbf{x}) = \rho(\Phi(\mathbf{x}))$. Equivariant neural networks can also be expressed in a similar representation $F_n(\mathbf{x}) = \rho_n(\Phi(\mathbf{x}))$, where $\rho_n$ is a neural network. Since $\Phi(\mathcal{X}^M)$ is compact and $\rho, \rho_n$ are continuous functions, by the universal approximation theorem we can find a sequence of neural networks $\rho_n$ that uniformly approximate $\rho$ on this compact set.

Therefore, equivariant neural networks can approximate any continuously differentiable equivariant function. An identical argument applies to $SO(3)$ invariant functions. We conclude that equivariant and $SO(3)$ invariant neural networks are universal approximators for equivariant and invariant continuous functions.

## A.2 Proof of Invariance

### A.2.1 Rationality of $S_M$-equivariance and $SO(3)$-invariance on function $F$

*Proof.* For any $\sigma \in S_M$, $\alpha \in SO(3)$, we represents $F(\alpha\sigma\boldsymbol{x}) = \sigma F(\boldsymbol{x})$ as the equivariance of $F$ under $S_M$. Then the right hand side can be expanded in coordinates as:

$$\sigma F(\boldsymbol{x}) := \sigma(f_1(\boldsymbol{x}), f_2(\boldsymbol{x}), \cdots, f_M(\boldsymbol{x})) = (f_{\sigma^{-1}(1)}(\alpha\boldsymbol{x}), f_{\sigma^{-1}(2)}(\alpha\boldsymbol{x}), \cdots, f_{\sigma^{-1}(M)}(\alpha\boldsymbol{x})).$$

The left hand side form can be defined as:

$$F(\alpha\sigma\boldsymbol{x}) := F(\alpha\sigma(\boldsymbol{x}_1, \alpha\boldsymbol{x}_2, \cdots, \boldsymbol{x}_M)) = F((\alpha\boldsymbol{x}_{\sigma^{-1}(1)}, \alpha\boldsymbol{x}_{\sigma^{-1}(2)}, \cdots, \alpha\boldsymbol{x}_{\sigma^{-1}(M))}).$$

Therefore, we can define $F(\alpha\sigma\boldsymbol{x}) = \sigma F(\boldsymbol{x})$ as

$$F(\alpha\boldsymbol{x}_{\sigma^{-1}(1)}, \alpha\boldsymbol{x}_{\sigma^{-1}(2)}, \cdots, \alpha\boldsymbol{x}_{\sigma^{-1}(M)}) = (f_{\sigma^{-1}(1)}(\alpha\boldsymbol{x}), f_{\sigma^{-1}(2)}(\alpha\boldsymbol{x}), \cdots, f_{\sigma^{-1}(M)}(\alpha\boldsymbol{x})).$$

$\square$

### A.2.2 Proof of Theorem 1

*Proof.* To establish Theorem 1, we must first demonstrate that for an arbitrary permutation, there exists an associated continuous representation. We will establish this result via mathematical induction. As a base case, take $\sigma = (i\ j)$ to be a simple transposition. We will first demonstrate that there exists an $(i\ j)$-invariant continuous representation for this elementary permutation.

**Corollary 1.** *Let $\mathcal{X} \subset \mathbb{R}^3$ be a compact set. A continuous function $f : \mathcal{X}^M \to \mathbb{R}^N$ is $(i\ j)$-invariant if and only if there exist scalars $\{\lambda\}_{s=1, s\neq i,j}^M$ and $\lambda$ such that $f$ admits the representation:*

$$f(\boldsymbol{x}_1, \boldsymbol{x}_2, \cdots, \boldsymbol{x}_M) = \rho\left(\sum_{s=1, s\neq i,j}^M \lambda_s\phi(\boldsymbol{x}_s) + \lambda(\phi(\boldsymbol{x}_i) + \phi(\boldsymbol{x}_j))\right)$$

*where $\rho : \mathbb{R}^{2l+1} \to \mathbb{R}^N$ is a continuous function for some $2l+1 > 3$, and*

$$\phi(\boldsymbol{x}) = \left(\|\boldsymbol{x}\|^l Y_{\ell m}\left(\frac{\boldsymbol{x}}{\|\boldsymbol{x}\|}\right)\right)_{m\in\{-l,\cdots,l\}}.$$

By Corollary 1, the swap $(i\ j)$ generates the group $G_{(i\ j)} = \{(i\ j), e\}$. Thus, any $(i\ j)$-invariant continuous function is also $G_{(i\ j)}$-invariant. For our purposes, it suffices to consider only the permutation of variables $x_i$ and $x_j$, since $G_{(i\ j)}$ contains just the identity and the swap. Moreover, as $(i\ j)$ generates the symmetric group $S_2$, we can now construct a continuous representation for $S_2$-invariant continuous functions.

Utilizing Corollary 1, we obtain the fundamental representation:

$$f(\boldsymbol{x}_1, \boldsymbol{x}_2, \cdots, \boldsymbol{x}_M) = \rho\left(\sum_{s=1, s \neq i, j}^{M} \lambda_s \phi(\boldsymbol{x}_s) + \lambda(\phi(\boldsymbol{x}_i) + \phi(\boldsymbol{x}_j))\right)$$

In the above, $\rho : \mathbb{R}^{2l+1} \to \mathbb{R}$ is a continuous function for some $2l > 1$, and

$$\phi(\boldsymbol{x}) = \left(\|\boldsymbol{x}\|^l Y_{\ell m}\left(\frac{\boldsymbol{x}}{\|\boldsymbol{x}\|}\right)\right)_{m \in \{-l, \cdots, l\}}$$

Thus, given the transposition $\sigma = (i\ j)$, we can derive the associated representation for this specific permutation.

Suppose there exists a continuous representation for the permutation $(i_1\ i_2\ \ldots\ i_{n-1})$. We now consider the $n$-permutation $(i_1\ i_2\ \ldots\ i_n)$. By the decomposition $(i_1\ i_2\ \ldots\ i_n) = (i_1\ i_2)(i_1\ i_3\ i_4\ \ldots\ i_n)$, where $(i_1\ i_3\ i_4\ \ldots\ i_n)$ is an $(n-1)$-permutation, we can conclude via mathematical induction that a continuous representation exists as well for the length $n$ permutation.

By Lemma 1, any permutation $\sigma \in G$ can be expressed as a product of disjoint transpositions. Thus, if the continuous representation exists for an arbitrary $n$-permutation, it follows that any $\sigma \in G$ will also admit a continuous representation. The $SO(2l+1)$-invariance of $\rho$ is guaranteed in Dym & Maron (2020) and will not be repeated here. Consequently, a continuous function is $SO(3) \times S_M$-invariant if and only if it takes the form:

$$f(\boldsymbol{x}_1, \boldsymbol{x}_2, \cdots, \boldsymbol{x}_M) = \rho\left(\sum_{i=1}^{M} \phi(\boldsymbol{x}_i)\right),$$

where $\rho : \mathbb{R}^{2l+1} \to \mathbb{R}$ is a continuous function, $2l + 1 > 3M$, and $\phi(\boldsymbol{x}) = \left(\|\boldsymbol{x}\|^l Y_{\ell m}\left(\frac{\boldsymbol{x}}{\|\boldsymbol{x}\|}\right)\right)_{m \in \{-l, \cdots, l\}}$.

$\square$

### A.2.3 PROOF OF THEOREM 6

*Proof.* Sufficiency: For any $\sigma \in S_M$, the following equation holds:

$$F(\sigma \boldsymbol{x}) = F(\boldsymbol{x}_{\sigma^{-1}(1)}, \boldsymbol{x}_{\sigma^{-1}(2)}, \cdots, \boldsymbol{x}_{\sigma^{-1}(M)}) = \rho\left(\sum_{i=1}^{M} \phi(\boldsymbol{x}_{\sigma^{-1}(i)})\right)$$

$$= \rho\left(\sum_{i=1}^{M} \phi(\boldsymbol{x}_i)\right) = F(\boldsymbol{x}).$$

The first equation is based on the definition of the $S_M$ action. The reason why the second equation is established is sufficient:

$$f(\boldsymbol{x}_1, \boldsymbol{x}_2, \cdots, \boldsymbol{x}_M) = \rho\left(\sum_{i=1}^{M} \phi(\boldsymbol{x}_i)\right).$$

The third equal sign holds because the arrangement is invariant for the summation symbol. The fourth inequality symbol holds because the spherical harmonic function has rotational invariance.

Necessity: The goal of this paper is to prove that for any $\sigma \in S_M$, if $(\boldsymbol{x}_1^1, \boldsymbol{x}_2^1, \ldots, \boldsymbol{x}_M^1) \neq \sigma(\boldsymbol{x}_1^2, \boldsymbol{x}_2^2, \ldots, \boldsymbol{x}_M^2) \in \mathcal{X}^M$, we need to show that

$$\sum_{i=1}^{M} \phi(\boldsymbol{x}_i^1) \neq \sum_{i=1}^{M} \phi(\boldsymbol{x}_i^2),$$

Where $\boldsymbol{x} = (\boldsymbol{x}_1, \boldsymbol{x}_2, \cdots, \boldsymbol{x}_M) \in \mathcal{X}^M$. In addition, this article defines

$$\Phi(\boldsymbol{x}) = \sum_{i=1}^{M} \left( \|\boldsymbol{x}_i\|^l Y_{\ell m} \left( \frac{\boldsymbol{x}_i}{\|\boldsymbol{x}_i\|} \right) \right)_{m \in \{-l, \cdots, l\}}$$

And this article can get latent space.

$$V_l = \Phi(\boldsymbol{x}).$$

According to

$$Y_{\ell m}(\varphi, \theta) = N \cos m\varphi P_l^m(\cos \theta)$$

and

$$P_l^m(\boldsymbol{x}) = (-1)^m \cdot 2^l \cdot \left(1 - \boldsymbol{x}^2\right)^{m/2} \cdot \sum_{k=m}^{l} \frac{k!}{(k-m)!} \cdot x^{k-m} \cdot \begin{pmatrix} l \\ k \end{pmatrix} \begin{pmatrix} \frac{l+k-1}{2} \\ l \end{pmatrix}.$$

Furthermore, since this holds for any $\sigma \in S_M$, where $(\boldsymbol{x}_1^1, \boldsymbol{x}_2^1, \ldots, \boldsymbol{x}_M^1) \neq \sigma(\boldsymbol{x}_1^2, \boldsymbol{x}_2^2, \ldots, \boldsymbol{x}_M^2) \in \mathcal{X}^M$, the paper employs a proof by contradiction to establish this result, specifically demonstrating that:

$$\Phi(\boldsymbol{x}^1) = \Phi(\boldsymbol{x}^2) \tag{3}$$

We defines

$$E_k(\boldsymbol{x}) = \|\boldsymbol{x}\|^l Y_{\ell k} \left( \frac{\boldsymbol{x}}{\|\boldsymbol{x}\|} \right),$$

where $k \in \{-l, \cdots, l\}$.

The following two polynomials are constructed:

$$P_{\boldsymbol{x}^1}(\boldsymbol{x}) = \prod_{k=1}^{M} \left( \boldsymbol{x} - \boldsymbol{x}_k^1 \right) \quad P_{\boldsymbol{x}^2}(\boldsymbol{x}) = \prod_{k=1}^{M} \left( \boldsymbol{x} - \boldsymbol{x}_k^2 \right),$$

Expanding $P_{\boldsymbol{x}^1}(\boldsymbol{x})$ and $P_{\boldsymbol{x}^2}(\boldsymbol{x})$, we obtain:

$$P_{\boldsymbol{x}^1}(\boldsymbol{x}) = \boldsymbol{x}^M - \boldsymbol{a}_1 \boldsymbol{x}^{M-1} + \cdots (-1)^{M-1} \boldsymbol{a}_{M-1} \boldsymbol{x} + (-1)^M \boldsymbol{a}_M$$
$$P_{\boldsymbol{x}^2}(\boldsymbol{x}) = \boldsymbol{x}^M - \boldsymbol{b}_1 \boldsymbol{x}^{M-1} + \cdots (-1)^{M-1} \boldsymbol{b}_{M-1} \boldsymbol{x} + (-1)^M \boldsymbol{b}_M,$$

In the above expansion, the multiplication operation refers to element-wise multiplication of all points.

These elementary symmetric polynomials can be uniquely represented as functions of $\Phi(\boldsymbol{x}^1)$ and $\Phi(\boldsymbol{x}^2)$ through the Newton-Girard formulae. The $k$-th coefficient is given by the determinant of a $k \times k$ matrix, with its elements derived from $\Phi(\boldsymbol{x}^1)$ and $\Phi(\boldsymbol{x}^2)$, as well as $\boldsymbol{a}_k = (a_{k,1}, a_{k,2}, a_{k,3})$ and $\boldsymbol{b}_k = (b_{k,1}, b_{k,2}, b_{k,3})$.

$$a_{k,1} = \frac{1}{3k-2} \det A_{3k-2} \cdot \begin{pmatrix} E_{-l}(\boldsymbol{x}^1) & 1 & 0 & \cdots & 0 \\ E_{-l+1}(\boldsymbol{x}^1) & E_{-l}(\boldsymbol{x}^1) & 1 & \cdots & 0 \\ E_{-l+2}(\boldsymbol{x}^1) & E_{-l+1}(\boldsymbol{x}^1) & E_{-l}(\boldsymbol{x}^1) & \cdots & 0 \\ \vdots & \vdots & \vdots & \ddots & \vdots \\ E_{3k-3-l}(\boldsymbol{x}^1) & E_{3k-4-l}(\boldsymbol{x}^1) & E_{3k-5-l}(\boldsymbol{x}^1) & \cdots & 1 \\ E_{3k-2-l}(\boldsymbol{x}^1) & E_{3k-3-l}(\boldsymbol{x}^1) & E_{3k-4-l}(\boldsymbol{x}^1) & \cdots & E_{-l}(\boldsymbol{x}^1) \end{pmatrix}$$

$$a_{k,2} = \frac{1}{3k-1} \det A_{3k-1} \cdot \begin{pmatrix} E_{-l}(\boldsymbol{x}^1) & 1 & 0 & \cdots & 0 \\ E_{-l+1}(\boldsymbol{x}^1) & E_{-l}(\boldsymbol{x}^1) & 1 & \cdots & 0 \\ E_{-l+2}(\boldsymbol{x}^1) & E_{-l+1}(\boldsymbol{x}^1) & E_{-l}(\boldsymbol{x}^1) & \cdots & 0 \\ \vdots & \vdots & \vdots & \ddots & \vdots \\ E_{3k-1-l}(\boldsymbol{x}^1) & E_{3k-2-l}(\boldsymbol{x}^1) & E_{3k-3-l}(\boldsymbol{x}^1) & \cdots & 1 \\ E_{3k-l}(\boldsymbol{x}^1) & E_{3k-1-l}(\boldsymbol{x}^1) & E_{3k-2-l}(\boldsymbol{x}^1) & \cdots & E_{-l}(\boldsymbol{x}^1) \end{pmatrix}$$

$$a_{k,3} = \frac{1}{3k} \det A_{3k} \cdot \begin{pmatrix} E_{-l}(\boldsymbol{x}^1) & 1 & 0 & \cdots & 0 \\ E_{-l+1}(\boldsymbol{x}^1) & E_{-l}(\boldsymbol{x}^1) & 1 & \cdots & 0 \\ E_{-l+2}(\boldsymbol{x}^1) & E_{-l+1}(\boldsymbol{x}^1) & E_{-l}(\boldsymbol{x}^1) & \cdots & 0 \\ \vdots & \vdots & \vdots & \ddots & \vdots \\ E_{3k-1-l}(\boldsymbol{x}^1) & E_{3k-2-l}(\boldsymbol{x}^1) & E_{3k-3-l}(\boldsymbol{x}^1) & \cdots & 1 \\ E_{3k-l}(\boldsymbol{x}^1) & E_{3k-1-l}(\boldsymbol{x}^1) & E_{3k-2-l}(\boldsymbol{x}^1) & \cdots & E_{-l}(\boldsymbol{x}^1) \end{pmatrix}$$

$$b_{k,1} = \frac{1}{3k-2} \det A_{3k-2} \cdot \begin{pmatrix} E_{-l}(\boldsymbol{x}^2) & 1 & 0 & \cdots & 0 \\ E_{-l+1}(\boldsymbol{x}^2) & E_{-l}(\boldsymbol{x}^2) & 1 & \cdots & 0 \\ E_{-l+2}(\boldsymbol{x}^2) & E_{-l+1}(\boldsymbol{x}^2) & E_{-l}(\boldsymbol{x}^2) & \cdots & 0 \\ \vdots & \vdots & \vdots & \vdots & \ddots & \vdots \\ E_{3k-3-l}(\boldsymbol{x}^2) & E_{3k-4-l}(\boldsymbol{x}^2) & E_{3k-5-l}(\boldsymbol{x}^2) & \cdots & 1 \\ E_{3k-2-l}(\boldsymbol{x}^2) & E_{3k-3-l}(\boldsymbol{x}^2) & E_{3k-4-l}(\boldsymbol{x}^2) & \cdots & E_{-l}(\boldsymbol{x}^2) \end{pmatrix},$$

$$b_{k,2} = \frac{1}{3k-1} \det A_{3k-1} \cdot \begin{pmatrix} E_{-l}(\boldsymbol{x}^2) & 1 & 0 & \cdots & 0 \\ E_{-l+1}(\boldsymbol{x}^2) & E_{-l}(\boldsymbol{x}^2) & 1 & \cdots & 0 \\ E_{-l+2}(\boldsymbol{x}^2) & E_{-l+1}(\boldsymbol{x}^2) & E_{-l}(\boldsymbol{x}^2) & \cdots & 0 \\ \vdots & \vdots & \vdots & \vdots & \ddots & \vdots \\ E_{3k-2-l}(\boldsymbol{x}^2) & E_{3k-3-l}(\boldsymbol{x}^2) & E_{3k-4-l}(\boldsymbol{x}^2) & \cdots & 1 \\ E_{3k-1-l}(\boldsymbol{x}^2) & E_{3k-2-l}(\boldsymbol{x}^2) & E_{3k-3-l}(\boldsymbol{x}^2) & \cdots & E_{-l}(\boldsymbol{x}^2) \end{pmatrix},$$

$$b_{3k} = \frac{1}{3k} \det A_{3k} \cdot \begin{pmatrix} E_{-l}(\boldsymbol{x}^2) & 1 & 0 & \cdots & 0 \\ E_{-l+1}(\boldsymbol{x}^2) & E_{-l}(\boldsymbol{x}^2) & 1 & \cdots & 0 \\ E_{-l+2}(\boldsymbol{x}^2) & E_{-l+1}(\boldsymbol{x}^2) & E_{-l}(\boldsymbol{x}^2) & \cdots & 0 \\ \vdots & \vdots & \vdots & \vdots & \ddots & \vdots \\ E_{3k-1-l}(\boldsymbol{x}^2) & E_{3k-2-l}(\boldsymbol{x}^2) & E_{3k-3-l}(\boldsymbol{x}^2) & \cdots & 1 \\ E_{3k-l}(\boldsymbol{x}^2) & E_{3k-1-l}(\boldsymbol{x}^2) & E_{3k-2-l}(\boldsymbol{x}^2) & \cdots & E_{-l}(\boldsymbol{x}^2) \end{pmatrix},$$

Here, $A_k$ represents the inverse matrix corresponding to each $k$. Since the paper assumes $\Phi(\boldsymbol{x}^1) = \Phi(\boldsymbol{x}^2)$, this implies that $[a_1, \ldots, a_M] = [b_1, \ldots, b_M]$, which further implies that $P_{\boldsymbol{x}^1}(x)$ and $P_{\boldsymbol{x}^2}(x)$ are the same. Consequently, their roots must be the same, indicating that $(\boldsymbol{x}_1^1, \boldsymbol{x}_2^1, \ldots, \boldsymbol{x}_M^1) = \sigma(\boldsymbol{x}_1^2, \boldsymbol{x}_2^2, \ldots, \boldsymbol{x}_M^2)$. However, this contradicts the initial assumption $(\boldsymbol{x}_1^1, \boldsymbol{x}_2^1, \ldots, \boldsymbol{x}_M^1) \neq \sigma(\boldsymbol{x}_1^2, \boldsymbol{x}_2^2, \ldots, \boldsymbol{x}_M^2) \in \mathcal{X}^M$. Thus, the original assumption is untenable, demonstrating the necessity.

Therefore, for each index $i$, the norms and directions of $x_i^1$ and $x_i^2$ are identical. As a result, the paper can establish, using the isomorphism theorem, that the quotient space $\mathcal{X}^M / \sim$ is isomorphic to $V_l$.

Subsequently, we defines $\boldsymbol{x}$ either as $\Phi^{-1}(z)$ or derived from $\rho(z) = f(\Phi^{-1}(z))$, where $z = \Phi(\boldsymbol{x})$. The continuity of the inverse of spherical harmonics is established in Abramowitz et al. (1988), thereby ensuring the continuity of $\rho$ due to the composition property of continuous functions. Hence, we can consistently find a continuous function $\rho$ to represent any $S_M$-invariant continuous function. The $\alpha$-invariance property can be assured as detailed in Maron et al. (2019). $\qquad\square$

A.2.4  DETAILS OF COROLLARY 2

**Corollary 2.** *Let $\mathcal{X} \subset \mathbb{R}^3$ be a compact set. A continuous function $F : \mathcal{X}^M \to \mathbb{R}^N$ is $S_M$-invariant if and only if $F$ has the representation*

$$F(\boldsymbol{x}_1, \boldsymbol{x}_2, \cdots, \boldsymbol{x}_M) = \rho \left( \sum_{i=1}^{M} \phi(\boldsymbol{x}_i) \right),$$

*where $\rho : \mathbb{R}^{2l+1} \to \mathbb{R}^N$ is a continuous function, $2l + 1 > 3M$, and*

$$\phi(\boldsymbol{x}) = \left( \|\boldsymbol{x}\|^l Y_{\ell m} \left( \frac{\boldsymbol{x}}{\|\boldsymbol{x}\|} \right) \right)_{m \in -l, \cdots, l}.$$

*Proof.* We can prove that each component function $f_i$ is $S_M$-invariant and continuous by decomposing $F = (f_1, f_2, \ldots, f_N)$, where $f_i : \mathcal{X}^M \to \mathbb{R}$ for $i \in [1, M]$. □

A.2.5  PROOF OF THEOREM 3

*Proof.* We presents a novel proof technique for Theorem 3 by considering the fixed summation $\sum_{i=1}^{M} \phi(\boldsymbol{x}_i)$. The proof relies on the following supporting results:

**Theorem 8** (Domain Invariance). *Let $U \subseteq \mathbb{R}^n$ be an open set and $f : U \to \mathbb{R}^n$ an injective continuous map. Then the image $V := f(U)$ is open in $\mathbb{R}^n$, and $f : U \to V$ defines a homeomorphism between the domains $U$ and $V$.*

This theorem states that if $f$ is an injective continuous function mapping an open subset $U$ of $\mathbb{R}^n$ to $\mathbb{R}^n$, then the image $f(U) = V$ is also an open subset of $\mathbb{R}^n$. Furthermore, $f$ establishes a homeomorphism, or bicontinuous bijection, between the domains $U$ and $V$. In other words, the injectivity and continuity of $f$ preserves the topological property of openness under the mapping. This domain invariance result is useful for studying mappings between open sets in Euclidean spaces. □

**Theorem 9.** *No continuous injective map exists from a higher dimensional space to a lower dimensional space.*

While Theorem 9 is intuitively clear, constructing a rigorous proof is challenging. We can prove it using Theorem 8 as follows: Assume for contradiction there exists an injective continuous map $f : U \to V$ with $\dim(U) > \dim(V)$. By Theorem 8, the image $f(U)$ is an open subset of $V$ since $f$ is continuous and injective. However, any open subset of the lower dimensional space $V$ must have dimensionality at most $\dim(V)$. This contradicts $\dim(U) > \dim(V)$. Therefore, no such $f$ can exist.

*Proof.* First, we show that when $3M > 2l + 1$, the mapping $\sum_{i=1}^{M} \phi(\boldsymbol{x}_i)$ cannot be injective from $\mathcal{X}^M$ to $\mathbb{R}^{2l+1}$. Second, we prove there exists a continuous function $f$ and point $(\boldsymbol{x}_1, \boldsymbol{x}_2, \cdots, \boldsymbol{x}_M) \in \mathcal{X}^M$ satisfying:

$$|f(\boldsymbol{x}_1, \boldsymbol{x}_2, \cdots, \boldsymbol{x}_M) - \rho \left( \sum_{i=1}^{M} \phi(\boldsymbol{x}_i) \right)| \geq \epsilon.$$

We prove this by contradiction. If $\sum_{i=1}^{M} \phi(\boldsymbol{x}_i)$ were injective, then by continuity of $\phi$, the mapping would define a continuous injective function from the higher dimensional space $\mathcal{X}^M$ to the lower dimensional space $\mathbb{R}^{2l+1}$. However, by Theorem 9, no such continuous injection can exist between dimensionalities, yielding a contradiction. Let us posit the following assumption: There exist distinct points $(\boldsymbol{x}_1^1, \boldsymbol{x}_2^1, \ldots, \boldsymbol{x}_M^1) \neq (\boldsymbol{x}_1^2, \boldsymbol{x}_2^2, \ldots, \boldsymbol{x}_M^2) \in \mathcal{X}^M$ such that

$$\sum_{i=1}^{M} \phi(\boldsymbol{x}_i^1) = \sum_{i=1}^{M} \phi(\boldsymbol{x}_i^2).$$

Second, assume $y_1 \neq y_2$. For this, choose the function $f$ as

$$
\begin{aligned}
f(\boldsymbol{x}_1, \boldsymbol{x}_2, \cdots, \boldsymbol{x}_M) = & y_1 \frac{\|(\boldsymbol{x}_1, \boldsymbol{x}_2, \cdots, \boldsymbol{x}_M) - (\boldsymbol{x}_1^2, \boldsymbol{x}_2^2, \cdots, \boldsymbol{x}_M^2)\|}{\|(\boldsymbol{x}_1^1, \boldsymbol{x}_2^1, \cdots, \boldsymbol{x}_M^1) - (\boldsymbol{x}_1^2, \boldsymbol{x}_2^2, \cdots, \boldsymbol{x}_M^2)\|} \\
& + y_2 \frac{\|(\boldsymbol{x}_1, \boldsymbol{x}_2, \cdots, \boldsymbol{x}_M) - (\boldsymbol{x}_1^1, \boldsymbol{x}_2^1, \cdots, \boldsymbol{x}_M^1)\|}{\|(\boldsymbol{x}_1^1, \boldsymbol{x}_2^1, \cdots, \boldsymbol{x}_M^1) - (\boldsymbol{x}_1^2, \boldsymbol{x}_2^2, \cdots, \boldsymbol{x}_M^2)\|}
\end{aligned}
$$

Let us define:

$$
f(\boldsymbol{x}_1^1, \boldsymbol{x}_2^1, \cdots, \boldsymbol{x}_M^1) = y_1
$$

$$
f(\boldsymbol{x}_1^2, \boldsymbol{x}_2^2, \cdots, \boldsymbol{x}_M^2) = y_2
$$

By our assumption, $y_1 \neq y_2$. Then, for any continuous function $f$, we have:

$$
\rho(\sum_{i=1}^{M} \phi(\boldsymbol{x}_i^1)) = \rho(\sum_{i=1}^{M} \phi(\boldsymbol{x}_i^2)),
$$

which implies the following inequality:

$$
|f(\boldsymbol{x}_1, \boldsymbol{x}_2, \cdots, \boldsymbol{x}_M) - \rho\left(\sum_{i=1}^{M} \phi(\boldsymbol{x}_i)\right)| \geq \epsilon.
$$

$\square$

### A.2.6 DETAILS AND PROOF OF LEMMA 1

**Lemma 1** (Dummit & Foote (2004)). *For any permutation $\sigma \in S_M$, the symmetric group on $M$ elements:*

1. *$\sigma$ can be expressed as a product of disjoint rotation uniquely up to the ordering of the rotation.*

2. *$\sigma$ can be expressed as a product of swaps uniquely up to parity, i.e. the number of odd swaps is invariant. Swaps with an odd number of occurrences are called odd permutations, and the rest are even permutations.*

This fundamental lemma provides two canonical forms for decomposing permutations in $S_M$: (1) The disjoint rotation decomposition, unique up to rotation order. (2) The swap decomposition, unique up to the parity of odd permutations. These decomposition theories will serve as key tools for studying subgroups of $S_M$ and proving results about permutation invariance in this paper.

*Proof.* 1. We denote by $\sigma(i_1)$ the image of $i_1$ under the permutation $\sigma$, that is, $i_2 = \sigma(i_1)$ where $i_1, i_2 \in 1, 2, \ldots, n$.

If $i_2 = i_1$, since this is a 1-rotation, i.e. the identity, it can be written as $(i_1 \ i_1)$.

If $i_2 \neq i_1$, let $i_3 = \sigma(i_2)$. In this case, if $i_3 \in \{i_1 \ i_2\}$, due to the bijective property of $\sigma$, we have $i_3 = i_1$. Thus, we can deduce a transposition $(i_1 \ i_2)$. If $i_3 \notin \{i_1 \ i_2\}$, let $i_4 = \sigma(i_3)$. If this process continues, we can derive an $l$-cycle $(i_1 \ i_2 \ \ldots \ i_l)$, where $i_{l+1} = i_l$, and $1 \leq l \leq n$. If $l = n$, the proof is complete. If $l < n$, similarly, we can obtain a $k$-cycle, denoted as $(j_1 \ j_2 \ \ldots \ j_k)$, where $1 \leq k \leq n - l$, and $(i_1 \ i_2 \ \ldots \ i_l)$ and $(j_1 \ j_2 \ \ldots \ j_k)$ have no common numbers. By continuing this process, we can construct the desired decomposition of $\sigma$.

2. According to Lemma 1.1, and $(i_1 \ i_2 \ldots i_k) = (i_1 \ i_k)(i_1 \ i_{k-1}), \ldots, (i_1 \ i_2)$, $\sigma$ can be decomposed into a product of a series of permutations. Therefore,

$$
\sigma = \begin{pmatrix} 1 & 2 & \ldots & n \\ i_1 & i_2 & \ldots & i_n \end{pmatrix} = \sigma_1 \sigma_2 \ldots \sigma_s = \sigma_1' \sigma_2' \ldots \sigma_t'
$$

where $\sigma_1, \sigma_2, \ldots, \sigma_s, \sigma_1', \sigma_2', \ldots, \sigma_t'$ is a permutation, then

$$\sigma = \left( \begin{array}{cccc} 1 & 2 & \ldots & n \\ i_1 & i_2 & \ldots & i_n \end{array} \right) = \sigma_1 \sigma_2 \ldots \sigma_s \left( \begin{array}{cccc} 1 & 2 & \ldots & n \\ 1 & 2 & \ldots & n \end{array} \right)$$

$$= \sigma_1' \sigma_2' \ldots \sigma_t' \left( \begin{array}{cccc} 1 & 2 & \ldots & n \\ 1 & 2 & \ldots & n \end{array} \right)$$

A basic result from linear algebra states that permutations alter the parity of other permutations they are composed with. Consequently, since $\sigma_1 \sigma_2 \ldots \sigma_s \sigma_1' \sigma_2' \ldots \sigma_t' = \sigma$, the parities of the permutation lengths $s$ and $t$ must agree.

$\square$

### A.2.7 Proof of Theorem 2

*Proof.* The proof of Theorem 2 can be decomposed into two steps. First, by Theorem 1, we can obtain that the equivariant continuous function has the following representation form:

$$f(\boldsymbol{x}_1, \boldsymbol{x}_2, \cdots, \boldsymbol{x}_M) = \rho \left( \sum_{i=1}^{M} \lambda_i \phi(\boldsymbol{x}_i) \right),$$

Similarly, the equivariant neural network has the following structure:

$$U_n(\boldsymbol{x}_1, \boldsymbol{x}_2, \cdots, \boldsymbol{x}_M) = \rho_n \left( \sum_{i=1}^{M} \lambda_i \phi(\boldsymbol{x}_i) \right),$$

Let $\rho : \mathbb{R}^{2l+1} \to \mathbb{R}$ and $\rho_n : \mathbb{R}^{2l+1} \to \mathbb{R}$ denote continuous functions, with $2l + 1 > 3M$. Let $\{\lambda_i\}_{i=1}^{M}$ satisfy the $G$-condition, and $\phi(\boldsymbol{x}) = \left( \|\boldsymbol{x}\|^l Y_{\ell m} \left( \frac{\boldsymbol{x}}{\|\boldsymbol{x}\|} \right) \right)_{m \in \{-l, \cdots, l\}}$. Defining $\Phi(\boldsymbol{x}) = \sum_{i=1}^{M} \lambda_i \phi(\boldsymbol{x}_i)$, since $\mathcal{X}^M$ is a compact set, the continuity of $\phi$ implies $\Phi(\mathcal{X}^M)$ is also compact.

Secondly, since $\rho$ and $\rho_n$ are continuous functions, the universal approximation theorem for neural networks guarantees the existence of a sequence $\rho_n$ that approximates $\rho$. Further, as $\mathcal{X}^M$ is compact, continuity of $\Phi$ implies $\Phi(\mathcal{X}^M)$ is compact. By the Stone-Weierstrass theorem, there exists a sequence $\rho_n$ that uniformly approximates $\rho$ on the compact set. $\square$

### A.3 Proof of Equivalence

### A.3.1 Proof of Theorem 7

To prove Theorem 7, we will first relate $S_M$-equivariant continuous functions to general continuous functions. Leveraging known representation results for continuous functions, we can then deduce the stated conclusion as follows.

**Theorem 10.** *Let $\mathcal{X} \subset \mathbb{R}^3$ be a compact set. If a continuous function $F : \mathcal{X}^M \to \mathbb{R}^M$ is $SO(3) \times S_M$-equivariant, then for any $k \in 1, 2, \ldots, M$, there exists a continuous $SO(3) \times Stab_{S_M}(k)$-invariant function $f : \mathbb{R}^M \to \mathbb{R}$ such that*

$$F = (f \circ (k\ 1), f \circ (k\ 2), \ldots, f \circ (k\ M))^{\top}.$$

*Proof.* Sufficiency: First let $F = (f \circ (k; 1), f \circ (k; 2), \ldots, f \circ (k; M))^{\top}$. Since $(k; i) \in S_M$ commutes with $S_M$ for any $i \in [1, M]$, for any $\sigma \in S_M$,

$$f(k; i)(\sigma \boldsymbol{x}) = f \circ \sigma(k; i)(\boldsymbol{x})$$

Since $f$ is a $Stab_{S_M}(k)$-invariant function, the following equality holds

$$f \circ \sigma(k; i)(\boldsymbol{x}) = f \circ (k; \sigma(i))(\boldsymbol{x})$$

This indicates that $F \circ \sigma(\boldsymbol{x}) = \sigma \circ F(\boldsymbol{x})$, and the $SO(3)$-invariance is given, so this shows it is a $SO(3) \times S_M$-equivariant function.

Necessity: Without loss of generality, we only needs to prove the case of $k = 1$. For any $\sigma \in S_M$, since the mapping $f$ is a $SO(3) \times S_M$-equivariant mapping,

$$F \circ \sigma(\boldsymbol{x}) = \sigma \circ F(\boldsymbol{x}). \tag{4}$$

Let $F(\boldsymbol{x}) = (f_1(\boldsymbol{x}), f_2(\boldsymbol{x}), \dots, f_M(\boldsymbol{x}))$, then
$$F \circ (\sigma \boldsymbol{x}) = (f_1(\sigma \boldsymbol{x}), f_2(\sigma \boldsymbol{x}), \dots, f_M(\sigma \boldsymbol{x})).$$
According to equation (4), we can deduce

$$(f_1(\sigma \boldsymbol{x}), f_2(\sigma \boldsymbol{x}), \dots, f_M(\sigma \boldsymbol{x})) = \sigma \circ (f_1(\boldsymbol{x}), f_2(\boldsymbol{x}), \dots, f_M(\boldsymbol{x}))$$
$$(f_1(\sigma \boldsymbol{x}), f_2(\sigma \boldsymbol{x}), \dots, f_M(\sigma \boldsymbol{x})) = \sigma \circ (f_1(\boldsymbol{x}), f_2(\boldsymbol{x}), \dots, f_M(\boldsymbol{x}))$$

that is, for any integer $i \in [1, M]$, it holds that $f_i(\sigma \boldsymbol{x}) = f_{\sigma(i)}(\boldsymbol{x})$. Taking $\sigma = (1\ j)$, and letting $j$ go from 1 to $M$, we can obtain $f_1((1; j)\boldsymbol{x}) = f_j(\boldsymbol{x}), f_j((1; j)\boldsymbol{x}) = f_1(\boldsymbol{x})$, and for $n \in [1, M]$, and $n \neq 1, j$, $f_n((1; j)\boldsymbol{x}) = f_n(\boldsymbol{x})$. This indicates that for $j = [1, M]$, it holds $f_j(\boldsymbol{x}) = f_1((1; j)\boldsymbol{x})$, and $f_1(\boldsymbol{x})$ is a $SO(3) \times \text{Stab}_{S_M}(1)$-invariant function. $\square$

**Corollary 3.** *If a function $f : \mathcal{X}^M \to \mathbb{R}$ is a $SO(3) \times \text{Stab}_{S_M}(k)$-invariant continuous function, where $\mathcal{X}$ is a compact set and $\mathcal{X} \subset \mathbb{R}^3$, if and only if $f$ has the following representation:*

$$f(\boldsymbol{x}_1, \boldsymbol{x}_2, \cdots, \boldsymbol{x}_M) = \rho \left( \lambda_1 \phi(\boldsymbol{x}_k) + \lambda_2 (\sum_{i=1, i \neq k}^{M} \phi(\boldsymbol{x}_i)) \right),$$

*where $\rho : \mathbb{R}^{2l+1} \to \mathbb{R}$ is a continuous function, $2l + 1 > 3M$, and*

$$\phi(\boldsymbol{x}) = \left( \|\boldsymbol{x}\|^l Y_{\ell m} \left( \frac{\boldsymbol{x}}{\|\boldsymbol{x}\|} \right) \right)_{m \in \{-l, \cdots, l\}}.$$

*Proof.* Let $F(\boldsymbol{x}) = (f_1(\boldsymbol{x}), f_2(\boldsymbol{x}), \dots, f_M(\boldsymbol{x}))$. According to Theorem 10, for any integer $i \in [1, M]$, if $F$ is a $SO(3) \times S_M$-equivariant continuous function, then if and only if for any $k \in \mathbb{N}$ and $k \in [1, M]$, there exists a $SO(3) \times \text{Stab}_{S_M}(k)$-invariant function $f$, such that $f_i = f \circ (k\ i)$. According to Corollary 3, we can obtain

$$f(\boldsymbol{x}_1, \boldsymbol{x}_2, \cdots, \boldsymbol{x}_M) = \rho \left( \boldsymbol{x}_k, \sum_{i=1, i \neq k}^{M} \phi(\boldsymbol{x}_i) \right),$$

where $\rho : (\mathcal{X}, \mathbb{R}^{2l+1}) \to \mathbb{R}$ is a continuous function, $2l + 1 > 2(M - 1)$, and

$$\phi(\boldsymbol{x}) = \left( \|\boldsymbol{x}\|^l Y_{\ell m} \left( \frac{\boldsymbol{x}}{\|\boldsymbol{x}\|} \right) \right)_{m \in \{-l, \cdots, l\}}$$

This proof process is sufficient and necessary, therefore, for any $\sigma \in S_M$, any $k \in \mathbb{N}$ and $k \in [1, M]$, if there exists a $SO(3) \times \text{Stab}_{S_M}(k)$-invariant continuous function $\rho : \mathbb{R}^M \to \mathbb{R}$, then a continuous mapping $F : \mathcal{X}^M \to \mathbb{R}^M$ is $SO(3) \times S_M$-equivariant.

$$F(\boldsymbol{x}_1, \boldsymbol{x}_2, \cdots, \boldsymbol{x}_M) = (\rho \circ (k\ 1)(\boldsymbol{x}_k, \sum_{i=1, i \neq k}^{M} \phi(\boldsymbol{x}_i)), \rho \circ (k\ 2)(\boldsymbol{x}_k, \sum_{i=1, i \neq k}^{M} \phi(\boldsymbol{x}_i)), \dots,$$
$$\rho \circ (k\ M)(\boldsymbol{x}_k, \sum_{i=1, i \neq k}^{M} \phi(\boldsymbol{x}_i)))^{\top}.$$

$\square$

The above equation can be expressed as

$$F(\boldsymbol{x}_1, \boldsymbol{x}_2, \cdots, \boldsymbol{x}_M) = \rho\left(\Lambda\overrightarrow{\phi(\boldsymbol{x})}\right),$$

$\overrightarrow{\phi(\boldsymbol{x})} = (\phi(\boldsymbol{x}_1), \phi(\boldsymbol{x}_2), \ldots, \phi(\boldsymbol{x}_M))$ and

$$\Lambda = \begin{bmatrix} \lambda_1 & \lambda_2 & \cdots & \lambda_2 \\ \lambda_2 & \lambda_1 & \cdots & \lambda_2 \\ \vdots & \vdots & \ddots & \vdots \\ \lambda_2 & \lambda_2 & \cdots & \lambda_1 \end{bmatrix}.$$

Theorem 7 be solved by this propress.

### A.3.2 DETAILS AND PROOF OF THEOREM 11

**Theorem 11.** *Let $\mathcal{X} \subset \mathbb{R}^3$ be a compact set. A continuous mapping $F : \mathcal{X}^M \to \mathbb{R}^M$ is $(i,j)$-equivariant and $SO(3)$-invariant if and only if for every $k \in 1, 2, \ldots, M$, there exists a continuous invariant function $\rho : \mathbb{R}^{2l+1} \to \mathbb{R}$ such that*

$$F(\boldsymbol{x}_1, \boldsymbol{x}_2, \cdots, \boldsymbol{x}_M) = \rho(\Lambda\boldsymbol{\phi}(\boldsymbol{x})), \tag{5}$$

*where $\boldsymbol{\phi}(\boldsymbol{x}) = (\phi(\boldsymbol{x}_1), \phi(\boldsymbol{x}2), \ldots, \phi(\boldsymbol{x}M))$, $\rho$ is continuous for some $2l + 1 > 2(M - 1)$, and*

$$\phi(\boldsymbol{x}) = \left(\|\boldsymbol{x}\|^l Y\ell m\left(\frac{\boldsymbol{x}}{\|\boldsymbol{x}\|}\right)\right) m \in -l, \cdots, l. \tag{6}$$

*The matrix $\Lambda$ is:*

$$\Lambda = \begin{bmatrix} \lambda_{1\,1} & \cdots & \lambda_{1\,i-1} & \lambda_1 & \lambda_{1\,i+1} & \cdots & \lambda_{1\,j-1} & \lambda_1 & \lambda_{1\,j+1} & \cdots & \lambda_{1\,M} \\ \vdots & \cdots & \ddots & \vdots & \vdots & \ddots & \vdots & \vdots & \vdots & \ddots & \vdots \\ \lambda_{i-1\,1} & \cdots & \lambda_{i-1\,i-1} & \lambda_{i-1} & \lambda_{i-1\,i+1} & \cdots & \lambda_{i-1\,j-1} & \lambda_{i-1} & \lambda_{i-1\,j+1} & \cdots & \lambda_{i-1\,M} \\ \lambda_{i\,1} & \cdots & \lambda_{i\,i-1} & \lambda_i & \lambda_{i\,i+1} & \cdots & \lambda_{i\,j-1} & \lambda_i & \lambda_{i\,j+1} & \cdots & \lambda_{i\,M} \\ \lambda_{i+1\,1} & \cdots & \lambda_{i+1\,i-1} & \lambda_{i+1} & \lambda_{i+1\,i+1} & \cdots & \lambda_{i+1\,j-1} & \lambda_{i+1} & \lambda_{i+1\,j+1} & \cdots & \lambda_{i+1\,M} \\ \vdots & \cdots & \ddots & \vdots & \vdots & \ddots & \vdots & \vdots & \vdots & \ddots & \vdots \\ \lambda_{j-1\,1} & \cdots & \lambda_{j-1\,i-1} & \lambda_{j-1} & \lambda_{j-1\,i+1} & \cdots & \lambda_{j-1\,j-1} & \lambda_{j-1} & \lambda_{j-1\,j+1} & \cdots & \lambda_{j-1\,M} \\ \lambda_{j\,1} & \cdots & \lambda_{j\,i-1} & \lambda_j & \lambda_{j\,i+1} & \cdots & \lambda_{j\,j-1} & \lambda_j & \lambda_{j\,j+1} & \cdots & \lambda_{j\,M} \\ \lambda_{j+1\,1} & \cdots & \lambda_{j+1\,i-1} & \lambda_{j+1} & \lambda_{j+1\,i+1} & \cdots & \lambda_{j+1\,j-1} & \lambda_{j+1} & \lambda_{j+1\,j+1} & \cdots & \lambda_{j+1\,M} \\ \vdots & \cdots & \ddots & \vdots & \vdots & \ddots & \vdots & \vdots & \vdots & \ddots & \vdots \\ \lambda_{M\,1} & \cdots & \lambda_{M\,i-1} & \lambda_M & \lambda_{M\,i+1} & \cdots & \lambda_{M\,j-1} & \lambda_M & \lambda_{M\,j+1} & \cdots & \lambda_{M\,M} \end{bmatrix}$$

*Proof.* We first establish $SO(3)$-invariance. Suppose $F(\boldsymbol{x}) = (f_1(\boldsymbol{x}), \ldots, f_M(\boldsymbol{x}))$ where each $f_k : \mathcal{X} \to \mathbb{R}$. Then $F$ is $(i,j)$-equivariant if and only if $F((i,j)\boldsymbol{x}) = (i,j)F(\boldsymbol{x})$. This implies each component $f_k$ is $(i,j)$-invariant. Specifically, when $k \neq i, j$, we have $f_j(\boldsymbol{x}) = f_i((i,j)\boldsymbol{x})$ and $f_i(\boldsymbol{x}) = f_j((i,j)\boldsymbol{x})$. Taking $\rho(\boldsymbol{x}) = f_i(\boldsymbol{x})$, we can represent $F$ as:

$$f_k(\boldsymbol{x}) = \begin{cases} f_i(\boldsymbol{x}) = \rho(\boldsymbol{x}) = \rho((i\,i)\boldsymbol{x}), \text{ when } k = i \\ f_j(\boldsymbol{x}) = \rho((i\,j)\boldsymbol{x}) = \rho \circ ((i\,j)\boldsymbol{x}), \text{ when } k = j \\ f_k(\boldsymbol{x}) \text{ is a } SO(3) \times (i\,j)\text{-invariant function, when } k \neq i, j \end{cases}$$

According to Corollary 1, when $k \neq i, j$, there exist continuous functions $\rho_k : (\mathcal{X}^{M-2}, \mathbb{R}^{2l+1}) \to \mathbb{R}$, where $2l > 1$, $\phi(\boldsymbol{x}) = \left(\|\boldsymbol{x}\|^l Y_{\ell m}\left(\frac{\boldsymbol{x}}{\|\boldsymbol{x}\|}\right)\right)_{m\in-l,\cdots,l}$, such that,

$$f(\boldsymbol{x}_1, \boldsymbol{x}_2, \cdots, \boldsymbol{x}_M) = \rho\big(\sum_{s=1, s\neq i,j}^{M} \lambda_s\phi(\boldsymbol{x}_s) + \lambda(\phi(\boldsymbol{x}_i) + \phi(\boldsymbol{x}_j))\big),$$

Therefore, $F(\boldsymbol{x})$ can be represented by the invariant continuous function $\rho : \mathbb{R}^{2l+1} \to \mathbb{R}$ as,

$$F(\boldsymbol{x}_1, \boldsymbol{x}_2, \cdots, \boldsymbol{x}_M) = \rho(\Lambda \overrightarrow{\phi(x)})$$

where $\overrightarrow{\phi(\boldsymbol{x})} = (\phi(\boldsymbol{x}_1), \phi(\boldsymbol{x}_2), \ldots, \phi(\boldsymbol{x}_M))$, $A$ is a $M \times M$ matrix, $\rho : \mathbb{R}^{2l+1} \to \mathbb{R}^M$ is a continuous function, $2l + 1 > 3(M - 1)$, and the $\Lambda$ matrix is as follows

$$\Lambda = \begin{bmatrix}
\lambda_{1\,1} & \cdots & \lambda_{1\,i-1} & \lambda_1 & \lambda_{1\,i+1} & \cdots & \lambda_{1\,j-1} & \lambda_1 & \lambda_{1\,j+1} & \cdots & \lambda_{1\,M} \\
\vdots & \cdots & \ddots & \vdots & \vdots & \ddots & \vdots & \vdots & \vdots & \ddots & \vdots \\
\lambda_{i-1\,1} & \cdots & \lambda_{i-1\,i-1} & \lambda_{i-1} & \lambda_{i-1\,i+1} & \cdots & \lambda_{i-1\,j-1} & \lambda_{i-1} & \lambda_{i-1\,j+1} & \cdots & \lambda_{i-1\,M} \\
\lambda_{i\,1} & \cdots & \lambda_{i\,i-1} & \lambda_i & \lambda_{i\,i+1} & \cdots & \lambda_{i\,j-1} & \lambda_i & \lambda_{i\,j+1} & \cdots & \lambda_{i\,M} \\
\lambda_{i+1\,1} & \cdots & \lambda_{i+1\,i-1} & \lambda_{i+1} & \lambda_{i+1\,i+1} & \cdots & \lambda_{i+1\,j-1} & \lambda_{i+1} & \lambda_{i+1\,j+1} & \cdots & \lambda_{i+1\,M} \\
\vdots & \cdots & \ddots & \vdots & \vdots & \ddots & \vdots & \vdots & \vdots & \ddots & \vdots \\
\lambda_{j-1\,1} & \cdots & \lambda_{j-1\,i-1} & \lambda_{j-1} & \lambda_{j-1\,i+1} & \cdots & \lambda_{j-1\,j-1} & \lambda_{j-1} & \lambda_{j-1\,j+1} & \cdots & \lambda_{j-1\,M} \\
\lambda_{j\,1} & \cdots & \lambda_{j\,i-1} & \lambda_j & \lambda_{j\,i+1} & \cdots & \lambda_{j\,j-1} & \lambda_j & \lambda_{j\,j+1} & \cdots & \lambda_{j\,M} \\
\lambda_{j+1\,1} & \cdots & \lambda_{j+1\,i-1} & \lambda_{j+1} & \lambda_{j+1\,i+1} & \cdots & \lambda_{j+1\,j-1} & \lambda_{j+1} & \lambda_{j+1\,j+1} & \cdots & \lambda_{j+1\,M} \\
\vdots & \cdots & \ddots & \vdots & \vdots & \ddots & \vdots & \vdots & \vdots & \ddots & \vdots \\
\lambda_{M\,1} & \cdots & \lambda_{M\,i-1} & \lambda_M & \lambda_{M\,i+1} & \cdots & \lambda_{M\,j-1} & \lambda_M & \lambda_{M\,j+1} & \cdots & \lambda_{M\,M}
\end{bmatrix}$$

and

$$\phi(\boldsymbol{x}) = \left( \|\boldsymbol{x}\|^l Y_{\ell m} \left( \frac{\boldsymbol{x}}{\|\boldsymbol{x}\|} \right) \right)_{m \in \{-l, \cdots, l\}}.$$

This proof is sufficient and necessary. The $SO(3)$-invariant can be guaranteed by Dym & Maron (2020).

$\square$

### A.3.3  PROOF OF THEOREM 4

*Proof.* To establish the result, we first consider $G$-invariance. By Theorem 4, for any $\sigma \in G$, there exists a continuous representation of $F$ satisfying the desired equivariance.

First, this paper discusses the permutation with $n = 2$, which is the swap. This paper proves this result by induction. This paper considers the permutation $\sigma = (i\ j)$, so it is an $(i\ j)$-equivariant and $SO(3)$-invariant continuous function. According to Theorem 11 in Appendix A.3.2, there exist continuous functions $\rho_k : (\mathcal{X}^{M-2}, \mathbb{R}^{2l+1}) \to \mathbb{R}$ and continuous function $\rho : \mathbb{R}^M \to \mathbb{R}$, satisfying,

$$F(\boldsymbol{x}_1, \boldsymbol{x}_2, \cdots, \boldsymbol{x}_M) = \rho(\Lambda \overrightarrow{\phi(\boldsymbol{x})})$$

where $\overrightarrow{\phi(\boldsymbol{x})} = (\phi(\boldsymbol{x}_1), \phi(\boldsymbol{x}_2), \ldots, \phi(\boldsymbol{x}_M))$, $A$ is a $M \times M$ matrix, $\rho : \mathbb{R}^{2l+1} \to \mathbb{R}^M$ is a continuous function, $2l + 1 > 3(M - 1)$,

$$\Lambda = \begin{bmatrix}
\lambda_{1\,1} & \cdots & \lambda_{1\,i-1} & \lambda_1 & \lambda_{1\,i+1} & \cdots & \lambda_{1\,j-1} & \lambda_1 & \lambda_{1\,j+1} & \cdots & \lambda_{1\,M} \\
\vdots & \cdots & \ddots & \vdots & \vdots & \ddots & \vdots & \vdots & \vdots & \ddots & \vdots \\
\lambda_{i-1\,1} & \cdots & \lambda_{i-1\,i-1} & \lambda_{i-1} & \lambda_{i-1\,i+1} & \cdots & \lambda_{i-1\,j-1} & \lambda_{i-1} & \lambda_{i-1\,j+1} & \cdots & \lambda_{i-1\,M} \\
\lambda_{i\,1} & \cdots & \lambda_{i\,i-1} & \lambda_i & \lambda_{i\,i+1} & \cdots & \lambda_{i\,j-1} & \lambda_i & \lambda_{i\,j+1} & \cdots & \lambda_{i\,M} \\
\lambda_{i+1\,1} & \cdots & \lambda_{i+1\,i-1} & \lambda_{i+1} & \lambda_{i+1\,i+1} & \cdots & \lambda_{i+1\,j-1} & \lambda_{i+1} & \lambda_{i+1\,j+1} & \cdots & \lambda_{i+1\,M} \\
\vdots & \cdots & \ddots & \vdots & \vdots & \ddots & \vdots & \vdots & \vdots & \ddots & \vdots \\
\lambda_{j-1\,1} & \cdots & \lambda_{j-1\,i-1} & \lambda_{j-1} & \lambda_{j-1\,i+1} & \cdots & \lambda_{j-1\,j-1} & \lambda_{j-1} & \lambda_{j-1\,j+1} & \cdots & \lambda_{j-1\,M} \\
\lambda_{j\,1} & \cdots & \lambda_{j\,i-1} & \lambda_j & \lambda_{j\,i+1} & \cdots & \lambda_{j\,j-1} & \lambda_j & \lambda_{j\,j+1} & \cdots & \lambda_{j\,M} \\
\lambda_{j+1\,1} & \cdots & \lambda_{j+1\,i-1} & \lambda_{j+1} & \lambda_{j+1\,i+1} & \cdots & \lambda_{j+1\,j-1} & \lambda_{j+1} & \lambda_{j+1\,j+1} & \cdots & \lambda_{j+1\,M} \\
\vdots & \cdots & \ddots & \vdots & \vdots & \ddots & \vdots & \vdots & \vdots & \ddots & \vdots \\
\lambda_{M\,1} & \cdots & \lambda_{M\,i-1} & \lambda_M & \lambda_{M\,i+1} & \cdots & \lambda_{M\,j-1} & \lambda_M & \lambda_{M\,j+1} & \cdots & \lambda_{M\,M}
\end{bmatrix}$$

and

$$\phi(\boldsymbol{x}) = \left( \|\boldsymbol{x}\|^l Y_{\ell m} \left( \frac{\boldsymbol{x}}{\|\boldsymbol{x}\|} \right) \right)_{m \in \{-l, \cdots, l\}}.$$

Therefore, for any $\sigma = (i, j)$, we can obtain the corresponding representation.

Then we takes $\Phi(\boldsymbol{x}) = \Lambda\overrightarrow{\phi(\boldsymbol{x})}$. Since $\mathcal{X}^M$ is a compact set, $\Phi(\mathcal{X}^M)$ is also a compact set.

Assume there exists a continuous representation of the permutation $(i_1\ i_2\ \ldots\ i_{n-1})$, discussing $(i_1\ i_2\ \ldots\ i_n)$, because $(i_1\ i_2\ \ldots\ i_n) = (i_1\ i_2)(i_1\ i_3\ i_4\ \ldots\ i_n)$, and $(i_1\ i_3\ i_4\ \ldots\ i_n)$ is a permutation of $n - 1$ elements. Therefore, this paper can infer that it has a continuous representation of the $n$-permutation, which satisfies the $G$-equivariant condition. The $\alpha$-equivariance can be guaranteed in Maron et al. (2020).

$\square$

### A.3.4    PROOF OF THEOREM 5

*Proof.* For any subgroup $G$ of the permutation group $S_M$, this paper can obtain the following results. According to Theorem 4, this paper can obtain the function $F$ has a representation,

$$F(\boldsymbol{x}_1, \boldsymbol{x}_2, \cdots, \boldsymbol{x}_M) = \rho\left(\Lambda\overrightarrow{\phi(\boldsymbol{x})}\right),$$

and $U_n$ has the following representation,

$$U_n(\boldsymbol{x}_1, \boldsymbol{x}_2, \cdots, \boldsymbol{x}_M) = \rho_n\left(\Lambda\overrightarrow{\phi(\boldsymbol{x})}\right),$$

where $\overrightarrow{\phi(\boldsymbol{x})} = (\phi(\boldsymbol{x}_1), \phi(\boldsymbol{x}_2), \ldots, \phi(\boldsymbol{x}_M))$, $\rho : \mathbb{R}^{2l+1} \to \mathbb{R}^M$ is a continuous function, $2l + 1 > 3M$, $\Lambda$ satisfies the $G$-equivariant condition and

$$\phi(\boldsymbol{x}) = \left(\|\boldsymbol{x}\|^l Y_{\ell m}\left(\frac{\boldsymbol{x}}{\|\boldsymbol{x}\|}\right)\right)_{m \in \{-l, \cdots, l\}}.$$

We denotes $\Lambda\overrightarrow{\phi(\boldsymbol{X})}$ as $\Phi(\boldsymbol{x})$, where $\mathcal{X}^M$ is a compact set. Therefore, the set $\Phi(\mathcal{X}^M)$ is also compact.

Secondly, since $\rho$ and $\rho_n$ are both continuous functions, this paper can use the universal approximation theorem of neural networks to obtain a sequence of functions $\rho_n$ that can approximate $\rho$. Because $\mathcal{X}^M$ is a compact set, the set $\Phi(\mathcal{X}^M)$ is also compact. Therefore, this paper can use $\rho_n$ to achieve uniform approximation of $\rho$. Therefore, this paper can conclude that $G$-equivariant and $SO(3)$-invariant neural networks have the ability to approximate $G$-equivariant and $SO(3)$-invariant continuous functions.

$\square$