# OpenReview forum: "Uniform Approximation of Equivariant/Invariant Neural Networks"
_ICLR.cc/2024/Conference — ICLR 2024 Conference Withdrawn Submission_

### Official Review · Reviewer_JME2 · 2023-10-30

**Soundness:** 3 good
**Presentation:** 2 fair
**Contribution:** 2 fair
**Rating:** 3
**Confidence:** 3

**Summary:**

The paper is a theoretical paper on ability of G-equivariant and SO(3)-invariant neural networks to approximate any G-equivariant and SO(3)-invariant function (uniform approximation), for any G that is a subgroup of S_M (the symmetric group, or group of permutations).
It provides a constraint, namely that 2l+1>3M, for the guaranteed existence of this mapping: the latent space needs to have at least equal dimenson with the input space (otherwise, the feature map going from cartesian coordinates to a sum of spherical harmonics is non injective and uniform approximation is not guaranteed.)

**Strengths:**

Originality:

It deals with the combination of G-equivariant x SO(3)-invariant functions that need to be represented by a neural network, something that has never been done before (where G is a subgroup of S_M)

Quality:

proofs are preceded by a proof sketch, much more readable in general.

Clarity:

The paper has a clear introduction with some key definitions recalled for the unexperienced reader.

Significance:

See weakenesses. Nowadays the best GNNs for atomic/molecular/etc(3D data) representation are rotation-equivariant (not rotation-invariant).

It is not clear at all how the research applies to enlightening the reader on how to build (subgroup-of-S_M)-equivariant GNNs.

**Weaknesses:**

Significance:

The main theorem is, as explained in definition 2, limited to SO(3) invariance (not SO(3) equivariance).

This is unfortunate: the said equivariance is only that of the symmetric group S_M, combined with invariance wrt SO(3).
This limits the impact of the research.

Clarity/Significance:

There is no concusion about practical implications for GNN building: in which cases do we have universal approximation, and in which cases do we NOT have that guarantee ? (I detail my remark as a question, below).

This limits the impact of the research.

Clarity/Quality:

There is a number of typos, useless comments or missing explanations, that make it look like an unfinished (or rushed) paper.

This limits the quality of the presentation, and can make it hard to assess the soundness, at times.



Originality:

the work seems to be very close to prior works, simply combining them for a new case (G-equivariance x SO(3)-invariance).

Admittedly, the 2l+1>3M constraint is, to my knwoledge, new.

**Questions:**

General/important questions/comments:

The key input of the paper, in my opinion, is the observation (and proof) that when 2l+1<3M, a representation may not exist (formally, it is proven it does not always exist, but then one could be lucky to have a simple function to represent, for which it exists.)
In practice, it seems like for most practical cases, M being the number of node in the input graph, or at least the number of nodes in the neighborhood of a point, and l being quite limited, for memory and compute-time limitations reasons, this inequality will be verified. For instance even with l=2, with just 2 neighbors (M=2), we have 2l+1=5 < 3M = 6. For M=4 neighbors (or maybe we have to include the center node, so it's just 3 neighbors), we would need more than l=5 to guarantee the existence of a representation.



First, this is not discussed in the paper, which ends abruptly with theorem 5.

Second, I am not sure this is the correct interpretation: in practice, the first layer of a GNN (SO(3) invariant, in this case) will perform an update which sends the local node features $x_m \in R^3$ to some feature space $R^{2l+1}$. Seen globally, this mapping is a mapping from $R^{3M}$ (for M nodes, each having node features $x_m\in R^3$) to $R^{(2l+1)M}$.
The subsequent layers are mappings: $ R^{(2l_{n}+1)M} \to R^{(2l_{n+1}+1)M} $, if we index the layer number (network depth) with $n$.

Seen like this, intuitively, the mappings allow to represent any function, as soon as $l\geq 1$.

Then of course, I may be wrong because the update rull may kill information (kill injectivity).

I think the paper is oriented on the idea of graph classification, i.e. the whole input $\in R^{3M}$ is mapped to an invariant quantity ($\in R$ I would guess), through an equivariant step where we have features in $R^{2l+1$}$.

I may be wrong about the interpretation of the conclusion, but in any case, the paper should conclude about the impact of its main theorem(s) on the (un)ability of some GNN architectures to represent any function (for some given kinds of tasks).


A criticism of how the paper is built:

I like that key concepts are introduced in pages 2 and 4,5,6. However, I think it would make more sense to make a short appendix introducing these known concepts to the unexperienced reader (for self-sufficiency of the text), but instead include proof's sketches inside the main text, and ideally when possible, even more intuitive explanations of why the results are what they are, and also of their implications for building Neural Networks.
Currently, a lot of space is lost to these preliminaries (I knew most of that material), while the proofs and results still remain obscure to the new reader (I knew the preliminaries beforehand but have never read the papers working on uniform approximations proofs that are cited here... and I often have a hard time following the results/proofs anyway).

------

More detailed questions/remarks:

Abstract: state the you refer to permutation-equivariance (otherwise people may assume it's about SO(3)-equivariance or SE(3)-equivariance..).
This is importantly misleading (e.g: I would have probably not bid on this paper otherwise).


Figure 1 seems incomplete. There are no colors.

Summary at end of page 2 is too fast for me.

equation in page 3 refers to \phi(x), which is then said to take values in R^{2l+1}. It can be guessed from the notation with the index $m\in[-l,...,l]$, but saying explicitly that it is in R^{2l+1} could be helpful.


The equation in page 3 (not numbered) provides phi, and is said to be a representation of f, although it seems to me it is a representation (or encoding really, because there are no learnable parameters in phi) of x, not f.


page 3: recall what is Stab_G(k) please, or at least refer to the full name in English (not just the mathematical shorthand).


above definition 3: the fact you let SO(3)-equivariance for future work should be aknoledged from the very start (abstract and introduction).

page 5 : and is invariant to the
-> and is independent from the


property 1: you could recall the defintion of being orthogonal, applied to the Y_lm's

page6, after definition 6, just before theorem 1: "this theorem" -> "the following theorem"   (I was confused between def 6 and the following text). Or, simply remove this preamble to the theorem, since there is an explanatory paragraph, similar to this one, right after theorem 1


Still about this small remark, just before theorem 1: this is the first time (with definition 6) that you mention that
phi(x) takes values in R^{2l+1}.  I believe this should be stated earlier.

Question:

If I understand correctly, this construction is valid for any l, and in a NN, one would concatenate l=0,1,2,.. up to some l_max. Correct ?


It is very nice that an example is given in page 7, however, I would like to see what is the solution (the set of lambdas) for the case of G=S_M. I expect that then all lambdas must be equal, making the construction rather trivial.  [I later checked out the sketch of proof and realize I'm correct].

This special case should be mentionned in the main text.



So theorem 1, in the case of G=S_M, is a reformulation of the generic equivariant GNN message-building as proposed e.g. in Tensor Field Networks (Thomas et al 2018, eq (3), page 6), although here you are restrained to the SO(3)-invariant case. In their case, the G-invariance is provided by the summation denoted $b\in S$ (sum over the neighbors of a generic central node).



Thus, the conclusion of section 3.1  seems rather weak to me: I would not call the construction of theorem 1  "novel", as you write:  "novel representation from SO(3)xS_M invariant function (...)".

Or, you should specify that the only novelty is in generalizing from S_M to any of its subgroups ?

(Or, it could be I misunderstood the context, then please clarify).



Typo below theorem 2:

S_M is a sphere, G is ... -> S_M is the symmetric group (grp of permutations). G is ...



The comment after theorem 3 is useless, it just repeats
theorem 3. It's not the only case of non-informative post-theorem comment, but this one is especially obviously non informative. A setch proof or other context, relationship with GNN or other kind of comment (or nothing) would be more appropriate.

Just before section 4, you mention a theorem 4.1. Is that a theorem of Wagstaff (2021) or of Wagstaff (2019)?


Title of section 4 has a typo: G-invariant should read G-equivariant

This typo is repeated in the introductory text of sec 4. (before sec 4.1): indeed, theorem 5 is about G-equivariance, not G-invariance.

And again in the title of subsection 4.1

And again, the paper ends very abruptly on theorem 5, without looping back to practical considerations for GNNs dealing with large graphs M>>1 of 3D point cloud data.

---

### Official Review · Reviewer_i9to · 2023-10-31

**Soundness:** 1 poor
**Presentation:** 2 fair
**Contribution:** 2 fair
**Rating:** 3
**Confidence:** 5

**Summary:**

The paper proposes approximation results for functions invariant/equivariant to the action of $SO(3)\times G$ on $R^{3M} $where $G$ is a general subgroup of $S_M$

**Strengths:**

The authors seemed to have some interesting theoretical insights (e.g., the relationship between $\ell$ and $M$) which, after a very thorough rewriting, could lead to an interesting paper.

**Weaknesses:**

* Main weakness: I'm pretty sure Theorem 1, and thus also later theorems which are based on Theorem 1, are not correct.



* Many inaccuracies and non-standard phrasings.

These weaknesses will be discussed in detail below.

**Questions:**

1. I believe Theorem 1 is not correct.

* Here is a counter example:
Say we consider the space of M=3 points in $R^3$, and take $G$ to be the group of permutations with positive sign $A_3$. First I claim that whenever $\ell\geq 1$, if $\lambda_1,\lambda_2,\lambda_3$ satisfy the $G$ invariant condition, then $\lambda_1=\lambda_2=\lambda_3$. To see this note that in particular we have that for every permutation $\tau \in G$ and every $x_1,x_2,x_3$
$$\lambda_1x_1+\lambda_2x_2+\lambda_3x_3=\lambda_1x_{\tau(1)}+\lambda_2x_{\tau(2)}+\lambda_3x_{\tau(3)} $$
Choosing $x_2=0=x_3$ and running over the different possible permutations we obtain that $\lambda_1=\lambda_2=\lambda_3$.

We conclude that in the equality in Theorem 1, we claim that a function f which is G invariant is going to always be equal the the composition of $\rho$ with an $S_3$ invariant function, since $\lambda_1=\lambda_2=\lambda_3$. So to get a counter example it is sufficient to show that there exists a functions which is $G$ invariant but not $S_3$ invariant. An example of such a function is
$$f(x_1,x_2,x_3)=det([x_1,x_2,x_3]) $$
Note that this function is also invariant to multiplication of all points $x_i$ by elements in $SO(3)$.

* Another reason why I think Theorem 1 is unlikely: In general it is difficult to come up with a scheme that is injective on $R^M$ modul a general subgroup $G$ of $S_M$. For example the action of permutations in $S_m$ on graphs with $m$ vertices can be rephrased as the action of a subgroup $G\cong S_m $ of $S_M, M=m^2 $. Injectivity module this group (a,k,a, graph isomorphism) has no polynomial time solution to date.

* In terms of the proof of Theorem 1: I could not find a proof of corollary 1. Moreover in page 14 the paragraph starting with the word "Assume", and page 18 the paragraphs starting with "suppose"- it is not clear why the fact that the characterization works for a single transposition, and the fact that every permutation is a compostion of transpositions, implies that the characterization would work for the permtuation, and then also for all permutations in G simultaneously.

2. The characterization in Theorem 1 is probably true when $G=S_M$ and all $\lambda_i$ are the same. If we only restrict ourselves to this case, do you require that $\rho$ in Theorem 1 is invariant to the action of $SO(3)$ on the spherical harmonic space via Wigner matrices? This seems natural to require as otherwise the functions in the characterization will not be $SO(3)$ invariant. In Theorem 2, what do your networks $U_n$ look like and are they "Wigner-equivaraint"? I couldn't understand this from the paper.

 3. Remarks re the writing in the paper:

*  Abstract: I think you should clarify when you say equivariant, with respect to which action
*  Regarding ZZnet: why do you say it discusses general groups? Doesn't it discuss $SO(2)\times S_M$?
*  Page 4: "the activation functions..." in what sense are some properties of activations necessary and some are not?
* Just above definition 1. Multiple typos in the definition of the group action.
* Also the notation $\alpha \circ \sigma$ is not standard, I'd suggest $(\alpha,\sigma)_*$ (of course this is not an error or anything)
* The paragraph under property 3 sounds incorrect or very badly phrased.
* Definition 6: I would delete the first sentence "consider an arbitrary permutation $\sigma$ since below you write $\forall \sigma \in G$.
* In Definition 6 I believe $\Phi$ maps into $R^{2\ell+1}$ and not just $R$.

---

### Official Review · Reviewer_7B9b · 2023-11-06

**Soundness:** 2 fair
**Presentation:** 1 poor
**Contribution:** 2 fair
**Rating:** 3
**Confidence:** 4

**Summary:**

This theoretical paper studies the approximation power of equivariant neural networks. It deals with architectures that are invariant with respect to the rotation group $SO(3)$ and equivariant with respect to a subgroup of the symmetric group $S_M$.
Theorem 1 gives a characterization of invariant continuous functions. It is an extension of the standard result proved for deepset.
Building on this result, Theorem 2 shows that such functions can be approximated by neural networks with the same symmetries. Theorem 3 gives a lower bound on the size of latent space.
Similar results are obtained for equivariant continuous functions in Section 4.

**Strengths:**

These results can be seen as a natural extension of the original results for invariant/equivariant functions in the fundamental paper introducing deepest.

**Weaknesses:**

The authors only provide theoretical results, and it is hard to see how these results can translate into practical neural networks.

Even from a theoretical point of view, I do not understand the interest of Theorems 2 and 5 dealing with neural networks. The definition of a neural network given by the author is very general, and it can be any function with a finite number of parameters. It seems clear that such powerful NN can approximate invariant/equivariant functions. Results in the literature connecting NN to approximation of continuous invariant/equivariant functions deal with limited architectures of NN see for example: Azizian, Waiss, and Marc Lelarge. "Expressive power of invariant and equivariant graph neural networks." arXiv preprint arXiv:2006.15646 (2020). which shows that different architectures have different approximation capabilities. Since you are not restricting the architecture of the NN considered in your paper, I do not see first how they can be implemented and second their approximation power is clear from a theoretical point of view.

**Questions:**

This paper is clearly not ready for publication, the authors should correct typos and problems like sections 3 and 4 having the same title!

Using the same symbol $\sigma$ for the activation function and permutations is not appropriate.

Please define properly how the symmetric group acts on $\mathbb{R}^n$ before definition 2.

Please give references for properties 1,2,3. I do not understand the last sentence of Section 2. Pleas explain it.

In the appendix your corollary 1 on page 17 is crucial but I did not find proof for it. Please provide a proof.

---

### Official Review · Reviewer_pKT6 · 2023-11-07

**Soundness:** 2 fair
**Presentation:** 1 poor
**Contribution:** 2 fair
**Rating:** 3
**Confidence:** 4

**Summary:**

The paper tried to establish representations (based on spherical harmonics) and approximations for continuous functions on Euclidean 3D clouds of unordered points.

**Strengths:**

The authors should be highly praised for trying to prove mathematical results. Since the arguments are rather technical, a proper reviewing is more suitable for journals without strict deadlines.

**Weaknesses:**

The word "problem" never appears in the paper, though a rigorous and explicit problem statement might have helped to understand the unresolved challenges.

Quote: "Spherical harmonics are introduced in Blanco et al. (1997)."

Comment: Spherical harmonics were introduced by Pierre-Simon de Laplace in 1782, see https://en.wikipedia.org/wiki/Spherical_harmonics.

Definition 5 requires a supremum because a maximum may not be attained (even by a continuous function) on an arbitrary space X.

In Definition 6 and all relevant places later, the scalar function phi should have the index l. The norm of a vector x often has two different notations |x| and ||x||.

Quote: "Theorem 9. No continuous injective map exists from a higher dimensional space to a lower dimensional space."

Comment: This result is taught to undergraduates in a basic topology course, please find a reference to a textbook. The provided argument uses the invariance of dimension, which is basically the same claim, so the argument is circular.

Since proofs do not immediately follow statements, every keyword "Proof" in itallic should specify what theorem is actually proved.

Quotes: "The complete proof is relegated to Appendix A.2.3", "Further details can be found in Corollary 1 of Appendix A.2.2", "The complete verification is furnished in Appendix A.3.1".

Comment: The authors should avoid giving forward references to unstated results such as "Appealing to Lemma 1 (proof in Appendix A.2.6)" because the paper is expected to be read from the beginning to the end, not in the opposite direction starting from Appendix A.2.6.

"Lemma 1 (Dummit & Foote (2004))" seems to be another classical result, which was known for centuries, but is stated in a very confusing with undefined words such as "disjoint rotation", which probably means a cyclic permutation. The authors are encouraged to learn the classical concepts starting from https://en.wikipedia.org/wiki/Cyclic_permutation.

Quote: "The general group can be continuously represented by spherical harmonics through induction."

Comment: When using mathematical induction, follow the basic rules: explain the parameter, start from the induction base, state the induction assumption and prove the induction step.

Theorem 8 is not a claim but a classical definition of a homeomorphism, see https://en.wikipedia.org/wiki/Homeomorphism

The so-called universal approximation property is a weak form of completeness saying that an invariant of a point cloud (under actions of a given group such as SO(3) of rotations) should separate (distinguish between or be injective on) all equivalence classes of clouds.

For all clouds of unordered points (hence invariant under permutations) in any Euclidean dimension, this completeness was proved for the distance-based invariants by Widdowson et al (CVPR 2023), which are also Lipschitz continuous under perturbations of points and computable in polynomial time in the number of points for a fixed Euclidean dimension.

Hence studying approximation properties of neural networks makes sense now in more general cases such as non-Euclidean spaces or for other groups, which are more exotic than SE(n).

**Questions:**

In Theorem 4, should the function rho have values in R^m instead of R? Then a different notation for this vector-function will be more appropriate.

Definition 2 and 4 in section 2 introduce equivariant networks. The equivariance property is very weak because any linear combination of given point coordinates is equivariant. For example, the center of mass of a point cloud is equivariant but is not invariant under SE(3)-transformations (rigid motion). This and other non-invariants cannot distinguish rigid patterns of point clouds. What's the motivation to study equivariance instead of invariance?

Quote: "We will first demonstrate that there exists an (i,j)-invariant continuous representation for this elementary permutation. Corollary 1."

The word "corollary" means a consequence of an already proved result. If Corollary 1 was meant a lemma (an auxiliary statement), please write it as an independent claim outside a proof. Where is a proof of Corollary 1, which is an essential step (case) of Theorem 1?

Quote from the proof of Theorem 3: "the mapping would define a continuous injective function from the higher dimensional space X^3M to the lower dimensional space R^{2l+1}.

Question. If X=R^3 and M=2, the SO(3)-class of two points x_1,x_2 under SO(3)-rotations around the origin 0 is determined by three pairwise distances |x_1|,|x_2|, and |x_1-x_2| in the triangle 0,x_1,x_2, where the first two distances can be swapped, hence |x_1|+|x_2| and |x_1|^2+|x_2|^2 can be used as permutation-invariant functions.

Do the authors agree that the space of 2-point SO(3)-classes in this case has dimension 3, which is less than 3M=6? Then how does the dimensionality argument work in this case?

In Theorem 7 (appendix A.1.4), is the same symbol phi used for every coordinate function in the M-tuple and also for the full vector of these functions?

Quote: "For any sigma in S_M, we can verify and calculate: F(sigma x) = rho (Λambda phi (sigma x) ) = ..."

Question. Could you please give a full proof for this chain of equalities?

Quote: "Our proof of SO(3)-invariance critically builds upon and extends Theorem 2 from the seminal work of Zaheer et al. (2017)"

Question. Why was this paper called seminal? The fact that any permutation-invariant function of real numbers x_1,...,x_M (similarly to vector coordinates) can be expressed by symmetric polynomials x_1^k+...+x_M^k for powers k=1,...,M has been known since the 17th century and taught in elementary algebra even to first year undergraduates at proper universities, see the history at https://en.wikipedia.org/wiki/Galois_theory, while the "deep set" paper rephrases this result in various forms in Lemma 4, Lemma 6, and Theorem 7, which are stated as "new".

Theorem 8 (Kolmogorov–Arnold representation) is given in a weaker form with a reference to a book chapter, which doesn't state the actual theorem and doesn't even have a reference to the original result, see https://en.wikipedia.org/wiki/Kolmogorov–Arnold_representation_theorem.

Final Theorem 9 (as well as Theorem 2 in this submission) directly follows from the Weierstrass approximation theorem, which was proved in the 19th century, see https://en.wikipedia.org/wiki/Stone–Weierstrass_theorem. The classical results mentioned above can be indeed called seminal. Deeper learning of basic mathematics seems essential here.

Quote: "SO(3)-invariance is guaranteed in Dym & Maron (2020)"

Question. Could you please write a full proof of SO(3)-invariance by explicitly stating the results from Dym & Maron (2020)?

It seems that the paper was revised several times because the appendix starts with proof outlines, while further sections include more theorems such as Theorem 6 similar to Theorem 1 and new proofs. Does it make sense to write a new paper from scratch starting from a problem? Here is the key question is: what problem has been solved in the paper?

Quote: "we demonstrate that Φ(x) = sum over i of phi(x_i) constitutes an injective mapping"

Where exactly is this (probably the most important claim in the paper) actually proved?

Does the main result (Theorem 1 or 6) essentially claim that all SO(3)-classes of clouds of M unordered points can be distinguished by spherical harmonics (evaluated at certain points) up to a fixed degree 2l+1 > 3M? Would it be possible to more explicitly state and prove this result in the partial case for M=2 points, which should be easier to check than current pages 19-20 in the supplementary materials?

Quote: "the spherical harmonic function has rotational invariance"

Question. This might be the key confusion. The spherical harmonics are not invariant under SO(3)-rotations, see the simple formulae showing the clear dependence on spherical angles at https://en.wikipedia.org/wiki/Spherical_harmonics#List_of_spherical_harmonics.

The spherical harmonics generate a functional subspace that is invariant under SO(3)-rotations. So the whole subspace is invariant but the basis spherical harmonics are not invariant.

Similarly, the whole 3D Euclidean space is invariant under SO(3)-rotations but the individual basis vectors (1,0,0), (0,1,0), (0,0,1) are not invariant under SO(3)-rotations. Do you agree?